# Stability and Generalizability in SDE Diffusion Models with Measure-Preserving Dynamics

**Weitong Zhang**[1]     **Chengqi Zang**[2]     **Liu Li**[1]     **Sarah Cechnicka**[1]

**Cheng Ouyang**[1,3]        **Bernhard Kainz**[1,4]

[1] Imperial College London, UK, [2] University of Tokyo, JP, [3] University of Oxford, UK
[4]Friedrich-Alexander University Erlangen-Nürnberg, GER
`weitong.zhang20@imperial.ac.uk`

## Abstract

Inverse problems describe the process of estimating the causal factors from a set of measurements or data. Mapping of often incomplete or degraded data to parameters is ill-posed, thus data-driven iterative solutions are required, for example when reconstructing clean images from poor signals. Diffusion models have shown promise as potent generative tools for solving inverse problems due to their superior reconstruction quality and their compatibility with iterative solvers. However, most existing approaches are limited to linear inverse problems represented as Stochastic Differential Equations (SDEs). This simplification falls short of addressing the challenging nature of real-world problems, leading to amplified cumulative errors and biases. We provide an explanation for this gap through the lens of measure-preserving dynamics of Random Dynamical Systems (RDS) with which we analyse Temporal Distribution Discrepancy and thus introduce a theoretical framework based on RDS for SDE diffusion models. We uncover several strategies that inherently enhance the stability and generalizability of diffusion models for inverse problems and introduce a novel score-based diffusion framework, the **D**ynamics-aware **SDE D**iffusion **G**enerative **M**odel (D$^3$GM). The *Measure-preserving property* can return the degraded measurement to the original state despite complex degradation with the RDS concept of *stability*. Our extensive experimental results corroborate the effectiveness of D$^3$GM across multiple benchmarks including a prominent application for inverse problems, magnetic resonance imaging.

## 1 Introduction

Diffusion probabilistic models [53, 54, 52] have demonstrated impressive performance across various image generation tasks, primarily by modeling a diffusion process and then learning an associated reverse process. Among the many commonly used approaches [63], diffusion models that incorporate the concept of score functions [28, 55] can capture the intrinsic random fluctuations of the forward diffusion process, positioning them as a good choice for in-depth analysis. Score-based generative models (SGMs) entail gradually diffusing images towards a noise distribution, and then generating samples by chaining the score functions at decreasing noise levels with score-based sampling approaches. One such example of an SGM with a score-based sampling technique, known as score matching [54], has gained popularity for density estimation [16]. It employs methods such as

38th Conference on Neural Information Processing Systems (NeurIPS 2024).

Langevin dynamics [21, 43, 30] and SDEs [29, 37, 31, 71] to simulate the underlying probability distribution of training samples.

However, vanilla unconditional SGMs can be extended to inverse problems by leveraging an implicit prior distribution, based on the available counterpart measurement, subjected to corruption and/or noise. To this end *transitionary SGMs* enable an iterative recovery of the data from this noisy counterpart, instead of relying on Gaussian white noise as a starting point [66, 37, 39, 12, 20, 34].

Intuitively, leveraging priors and a generative capacity into transitionary SGMs offers the possibility of exploring high quality reconstruction and restoration and gaining better performance. However, current transitionary SGMs that incorporate priors have largely overlooked the unreliable quality of the prior and its measurement. Empirically, transitionary SGMs cannot always be trusted in terms of stability and efficiency, especially in a regime of non-uniformly distributed noise or corrupted signal quality [9, 66, 37]. Hence, the exploitation of transitional learning within SGMs does not come without costs as their advantages vanish in limited data quality settings. Theoretical understanding is notably lacking in this field with the following fundamental open problem: Can we realize reliable transitionary diffusion processes in practice for inverse problems with a theoretical guarantee?

While recent works have started to lay down a theoretical foundation for these models, a detailed understanding is still lacking. Current best practice advocates for smaller initialisation values (*e.g.*, noise schedule [66, 20], instead of large values [15, 61] to ensure that the forward dynamics brings the diffusion sufficiently close to a known prior and simple noise distribution. However, a proper choice of the values conditioned on the prior within a theoretical framework should be preferred for a better approximation of the score-matching objective and higher computational efficiency. To fully facilitate the power of reversion and generation of transitionary SGMs and to mitigate the influence of low-quality measurements for solving inverse problems, this paper provides a measure-preserving dynamics of random dynamical system (RDS) perspective as a promising way to obtain reliable reversion and generation. Notably, our 'measure' is not only the observations (*e.g.*, degraded images), but also represents the invariant probability measure (distribution) of the RDS. This allows to consider the concept of an RDS *stability* and to frame challenging degradation learning within a measure-preserving dynamical system. Thus, we can start from a transitionary SGM interpretation of diffusion models and connect RDS to the SDE in transitionary SGMs. The pitfalls (*e.g.,* Instability) are discussed in Sect. 3 and further implications can be found in the Appendix. Transitionary SGMs have not been fully explored before, and we provide a theoretical interpretation of a stationary process as a possible solution.

Our D$^3$GM framework is abstracted from transitionary SGMs. The key to our framework is a stationary process following measure-preserving dynamics to ensure the stability and generalizability of the diffusion, as well as reducing the influence of accumulated error, distribution bias and degradation discrepancy. Our contributions can be summarised as follows:

**1. Temporal Distribution Discrepancy**: We conduct a rigorous theoretical examination of the instability issue of transitionary SGMs, measured as Temporal Distribution Discrepancy (*i.e.*, lower bound of modeling error). This analysis sheds light on critical aspects related to stability and generalizability[1], effectively addressing an unexplored fundamental gap in the understanding of solving challenging inverse problems with SDEs.

**2. D$^3$GM Framework**: We propose a solution, D$^3$GM, and an explanation from measure-preserving dynamics of Random Dynamical Systems (RDS). 'Measure' includes both measurements (degraded image) and invariant measures (distribution) of RDS, which allows complex degradation learning and enhances restoration and reconstruction accuracy.

**3. Thorough Evaluation**: Our contributions are substantiated by extensive validation. We demonstrate the practical benefits of our D$^3$GM framework across various benchmarks, including challenging tasks such the reconstruction of Magnetic Resonance Imaging (MRI) data.

We address the instability of diffusion models for inverse problems under domain shift and concept drift (unknown and heterogeneous degradation). This leads to what we believe is a completely novel view on the theoretical foundation of how the *degradation* process is modelled. The result is an approach that is more in line with the original intention of the theory of diffusion. We

---

[1]Generalizability refers to the extent to which out-of-distribution and domain shift impacts the fidelity of the restoration process. Stability in SDE diffusion models is demonstrated by its resilience to degradation beyond its domain and its consistent ability to restore high-quality images.

chose inverse problems as a relevant application area to demonstrate our ideas but also included a variety of challenging problem settings to explore the generalizability of $D^3GM$. To the best of our knowledge, no other method can handle a **diverse range** of challenging tasks like *real-world* dehazing, *compressed* MRI reconstruction, *blind* MRI super-resolution, etc. with a **unified** underlying theoretical framework. In Tab. 1 we illustrate the key differences of $D^3GM$ compared with SGMs and transitionary SGMs.

## 2 Preliminaries

**SGMs.** We will follow the typical construction of the diffusion process $\boldsymbol{x}(t)$, $t \in [0, T]$ with $\boldsymbol{x}(t) \in \mathbb{R}^d$. Concretely, we want $\boldsymbol{x}(0) \sim p_0(\boldsymbol{x})$, where $p_0 = p_{\text{data}}$, and $\boldsymbol{x}(T) \sim p_T$, where $p_T$ is a tractable distribution that can be sampled. In this work, we consider the score-based diffusion form of the SDE [55]. Consider the following Itô diffusion process defined by an SDE:

$$d\boldsymbol{x} = \boldsymbol{f}(\boldsymbol{x}, t)dt + g(t)d\boldsymbol{W}, \tag{1}$$

where $\boldsymbol{f} : \mathbb{R}^d \mapsto \mathbb{R}^d$ is the drift coefficient of $\boldsymbol{x}(t)$, $g : \mathbb{R} \mapsto \mathbb{R}$ is the diffusion coefficient coupled with the standard d-dimensional Wiener process $\boldsymbol{w} \in \mathbb{R}^d$. By carefully choosing $\bar{f}, g$, one can achieve a spherical Gaussian distribution as $t \to T$. For the forward SDE in Eq. 1, there exists a corresponding reverse-time SDE [3, 55]:

$$d\boldsymbol{x} = [\boldsymbol{f}(\boldsymbol{x}, t) - g(t)^2 \underbrace{\nabla_{\boldsymbol{x}} \log p_t(\boldsymbol{x})}_{\text{score function}}]dt + g(t)d\boldsymbol{W}, \tag{2}$$

where $dt$ is the infinitesimal negative time step, and $\boldsymbol{w}$ is the Brownian motion running backwards. The score function $\nabla_{\boldsymbol{x}} \log p_t(\boldsymbol{x})$ is in general intractable and thus SDE-based diffusion models approximate it by training a time-dependent neural network under a score function [57, 28].

**Transitionary SGMs.** [17, 37, 66, 34, 20, 12] leverage a transitionary iterative denoising paradigm for the inverse problems. In inverse problems, such as super-resolution, we have an (nonlinear, partial, and noisy) observation $\boldsymbol{y}$ of the underlying high-quality signal $\boldsymbol{x}$. The mapping $\boldsymbol{x} \mapsto \boldsymbol{y}$ is many-to-one, posing an ill-posed problem. In this case, a strong prior on $\boldsymbol{x}$ is needed for finding a realistic solution. Formally, the general form of the *forward* (measurement) model is:

$$\boldsymbol{y} = \mathcal{A}(\boldsymbol{x}) + \boldsymbol{n}, \quad \boldsymbol{y}, \boldsymbol{n} \in \mathbb{R}^n, \boldsymbol{x} \in \mathbb{R}^d, \tag{3}$$

where $\mathcal{A}(\cdot)$: $\mathbb{R}^d \mapsto \mathbb{R}^{n2}$, oftentimes $n \ll d$ is the forward measurement operator and $\boldsymbol{n}$ is the measurement noise, assuming $\boldsymbol{n} \sim \mathcal{N}(0, \sigma^2 \boldsymbol{I})$. While sharing a similar aim of bridging $\boldsymbol{y}$ and $\boldsymbol{x}$ in transitionary SGMs, different mathematical frameworks have been used: [17] employs Inversion by Direct Iteration; [37, 34, 20] model it as a Mean-reverting SDE.

*Transitionary SGM* has become an increasingly important line of SDE research due to the applicability on images with theoretical guarantees. However, they often perfrom poorly in real-world scenarios. To provide a theoretical investigation of this gap, we interpret *Transitionary SGM* as *Ornstein-Uhlenbeck (OU)* process. This perspective allows us to understand the random fluctuations in image degradation as stochastic processes, providing a foundation to integrate *random dynamical systems (RDS)* with the diffusion process as a natural extension of the SDE framework involving the OU process.

The *Measure-preserving property* is introduced from the perspective of RDS: The distribution can still return to the original state despite severe degradation. Our approach constructs a bridge from measure-preserving dynamical system to transitionary SGM through measure-preserving dynamics, and highlights the *Temporal Distribution Discrepancy* in Sect. 3. Subsequently, we address this issue of instability: by incorporating a measure-preserving strategy into the solution of inverse problems, which is detailed in Sect. 4. This covers counterpart modeling, bridging a transition from uncertain diffusion modeling to deterministic solutions, yielding significant improvements in both performance and efficiency as demonstrated in Sect. 5. More details can be found in Sect. 6 and Appendix.

## 3 Instability Analysis: Transitionary SGMs with Corrupted SDE Diffusion

**Ornstein-Uhlenbeck (OU) process.** An OU process is a common case in transitionary SGMs, where $x_t$ is defined using an SDE: $dx_t = -\theta_t x_t dt + \sigma_t dW_t$. $W_t$ is standard Brownian motion. A drift term $\mu$ can be introduced:

---

[2]MRI signals are defined on $\mathbb{C}^n$ and $\mathbb{C}^d$. We demonstrate in Sect. 5 that our approach is applicable to MRI.

Table 1: Differences between state-of-the-art SDE diffusion-based approaches.

| Model | $p(X_0)$ | $p(X_1)$ | Theorem Foundation | Properties | TDD (Prop.2) | Attractor | Operator Dir. | Inverse problem solver |
|---|---|---|---|---|---|---|---|---|
| SGM [53] | $p_A$ | $\mathcal{N}(0, I)$ | SDE diffusion | *No prior* | Unsolvable | No subset attracts | one-sided | |
| IR-SDE [37] | $p_A$ | $p_B(\cdot\|X_0, \mu)$ | Mean-reverting SDE | *Instability* | Limited | Gaussian | one-sided | Lin/Non-Lin |
| I²SB [34] | $\delta_a$ | $p_B(\cdot\|X_0)$ | Schrodinger Bridge | *Strict Prior* | Limited | No subset attracts | one-sided | Lin/ Non-Lin |
| D³GM (ours) | $p_A$ | $p_B(\cdot\|X_0, \mu, \tau)$ | Measure-Preserving RDS | *Stability* | Robust | $\mathcal{N}(\mu, \tau^2\sigma^2 I)$ | two-sided | Lin/ Non-Lin/Blind |

$$dx_t = \theta_t(\mu - x_t)dt + \sigma_t dW_t, \tag{4}$$

where $\mu$ denotes state mean, reflecting the expected state of the measurement (*e.g.*, corrupted image [37], noisy speech [59]) over time. $\theta_t$, $\sigma_t$ are time-dependent parameters. The drift term corrects deviations from the constant $\mu$, effectively pulling the process towards $\mu$ ($t \to \infty$) with *Stability* (in Appx. G) as opposed to pure noise in Eq. 1.

**Measure-preserving Dynamics in SDE Diffusion.** The solution of the above SDE can be represented by a continuous-time random dynamical system $\varphi$ defined on a complete separable metric space $(X, d)$. (See precise definition of RDS in Appx. C). More generally, we can extend the RDS to a two-sided solution operator with a flow map. The base flow driven by Brownian motion can be written as $W(t, \vartheta_s(\omega)) = W(t + s, \omega) - W(s, \omega)$.

**Proposition 1** *After extending the solution of the OU process to RDS, the measure-preserving RDS $\varphi$ should meet the property $\varphi(t, s; \omega)x = \varphi(t - s, 0; \vartheta_s\omega)x$. However, OU processes with time-varying coefficients usually do not satisfy this property. In this situation, the system breaks the forward-reverse processes, making it difficult to maintain stability.*

> **Intuition 1.** A two-sided measure-preserving random dynamical (MP-RDS) system formulation enables us to use the Poincare recurrence theorem [45], (see precise statement in Appx.D), intuitively, with a two-sided MP-RDS $\varphi_t$, the Poincare recurrence theorem ensures that the system $\varphi_t$ starts from terminal condition $x_T$, run backward in time, will hit a region $(x_0 - \epsilon, x_0 + \epsilon)$ for small $\epsilon$ in finite time, where $x_0$ is the high-quality image.

> **Example 1.** Following Intuition 1. and Prop. 1, suppose that the OU's $\theta_t$ follows a cosine schedule, such that $\theta_t = \cos(t)$ for $0 \le t \le T$, then for some $0 \le s \le t \le T$, $\varphi(t, s; \omega)x \ne \varphi(t - s, 0; \vartheta_s\omega)x$ because the change of $\theta_t$ w.r.t. time is not uniform. The OU-process instability exists due to Temporal Distribution Discrepancy (Prop. 2).

At a high level, Proposition 1 can be extended to show that there exists a compact attracting set at any $-\infty < t < \infty$, and this convention has allowed us to characterize the attractor $K(\omega) = \mathcal{N}(\mu, \sigma^2/2\theta)$. The closed-form distribution for $\mathbf{y}$ can be complex and may not be tractable depending on the particular scenarios of the actual image degradation process $\boldsymbol{\mu}$. The modification of $\boldsymbol{\sigma}$ and $\boldsymbol{\theta}$ is used to regularize the perturbation and attempt to close the distribution. However, these injections might bypass the stationary process. More details can be found in Appx. D.

**Instability-Temporal Distribution Discrepancy.** Given the process OU $(x_t, \mu; t, \theta)$ with Eq. 4, where $x_T \ne x_\infty$ for finite $T$, indicates that the perturbed state cannot move towards the degraded LQ image and fails in matching the theoretical distribution. This inherent discrepancy further causes bias in the estimation of $\mu_t$, which gradually accumulates into error in the reverse process.

*Temporal Distribution Discrepancy* is illustrated by Proposition 2 (proof can be found in Appx. E):

**Proposition 2** *Given Eq. 3 and Eq. 4, and assume that the score function is bounded by $C$ in $L^2$ norm, then the discrepancy between the reference and the retrieved data is, with probability $(1 - \delta)$ at least:*

$$\begin{aligned}
\|x_0 - \mathrm{OU}(x_0, \mu; T, \theta)\|_2^2 \ge & \; | \left((x_0 - \mu)^2 - \sigma_T^2/2\theta_T\right) \mathrm{e}^{-2\bar{\theta}_T} + \sigma_T^2/2\theta_T \\
& - \sigma_{max}^2 \left(C\sigma_{max}^2 + d + 2\sqrt{-d \cdot \log\delta} - 2\log\delta\right) |,
\end{aligned} \tag{5}$$

> **Intuition 2.** Intuitively, Proposition 2 provides a theoretical measurement on how the difference between finite iteration distribution and the asymptotic distribution of the OU-process in $L^2$ could further enlarge the discrepancy between the retrieved image and the actual HQ image. Discrepancies are typically explained by complex degradation *vs.* monotonic modeling.

For a noisy inverse problem, the retrieved data with any finite $T$ depends on $\sigma_t, \mu_t, \lambda, T, \bar{K}$ (Lipschitz constant). This proposition also correlates to [39], where the lower bound of the distance between the high quality image and retrieved image in $L^2$ norm in our model is further enlarged by this discrepancy, which correlates to the term $\left((x_0 - \mu)^2 - (\sigma_T^2/2\theta_T)\right) e^{-2\bar{\theta}_T} + \sigma_T^2/2\theta_T$.

Another way to further minimize this bound is through the term $e^{-2\bar{\theta}_T}$ with $[0, T]$ normalized to $[0, 1]$. What we refer to as $\theta$-schedule corresponds to the exact functional form of $\theta_t$, several schedules can be set here, *e.g.*, constant, linear, cosine, and log. At a high level, the discrepancy between the reference and the retrieved data stems from the divergence between the forwarded final state and the low quality image. Eq. 5 can be factored into three constituent parts: the data residual, the stationary disturbance, and the random noise. While conditional diffusion generation entails a trade-off between variability and faithfulness [66], the persistent discrepancy within the residual has a significant impact on the generalizability of solving the transitionary tasks. This also establishes a connection with SDEdit [39] and CCDF [12]. When fitting inverse problems involving paired data into diffusion models, while accounting for deviations and degradation, inserting them into Eq. 4 directly may not be the most effective strategy.

During sampling and inference with the degraded input $y$, the discrepancy identified in Prop. 2 intensifies. The complex degradations in $y$ exacerbate the divergence from the expected $\mu$ distribution, significantly impacting the accuracy of the restored data $\hat{x}_0$. More details are in Appx. F.

# 4 Towards Stability: Measure-preserving Dynamics in SDE Diffusion

In Sect. 3, we extrapolate and theorize the Temporal Distribution Discrepancy on $\mu$ and $x$ in the diffusion model for the inverse problem. Our key idea is to combine the stationary process to alleviate the Temporal Distribution Discrepancy problem following the measure-preserving dynamics from RDS. Recall that our 'measure' is not only the measurements (*i.e.*, degraded image), but also represents the *invariant measure* (distribution) of the RDS.

We begin by describing the forward and reverse processes of the D³GM, which serves as a stable bridge between the quality data and the counterpart measurement. We adapt score-based training methods to estimate this SDE. Following this, we describe the essential constructions for preserving the stationary process in the diffusion model and solving for Temporal Distribution Discrepancy on an orthogonal basis compared to current transitionary diffusion models.

**Measure-preserving Dynamics with the Stationary Process.** Following Prop. 1, in a SDE Diffusion from $0$ to $T$, the corresponding 'attractors' (states) can be viewed as $\mathcal{N}\left(\mu + (x_0 - \mu)e^{-\bar{\theta}_t}, \sigma_t^2(1 - e^{-2\bar{\theta}_t})/2\theta_t\right)$. We can guide SDE Diffusion towards a stable and robust solution based on the properties of Measure-preserving in RDS. It can be extended to impose that for every $t$, $\frac{\sigma_t^2}{2\theta_t} = \lambda^2$, where $\lambda$ is the variance of the designated stationary measure forward process. This convention allows us to reduce the regularization on two variables $\sigma_t, \theta_t$ to just one variable to satisfy the property of the measure-preserving dynamics in the asymptotic sense, *i.e.*, $\lim_{t\to\infty} \varphi(t, s; \omega)x = \lim_{t\to\infty} \varphi(t - s, 0; \vartheta_s\omega) x$. This convention allows us to characterize the attractor of the system as $K(\omega) = \mathcal{N}(\mu, \lambda^2)$.

The definition and constraint of the attractor are significant; without imposing this constraint, the measure-preserving property cannot be maintained, and the system would degrade into a Coefficient Decoupled SDE (Coe. Dec. SDE), we also analyse this in Fig. 2 and Tab. 11 in Appx. H.

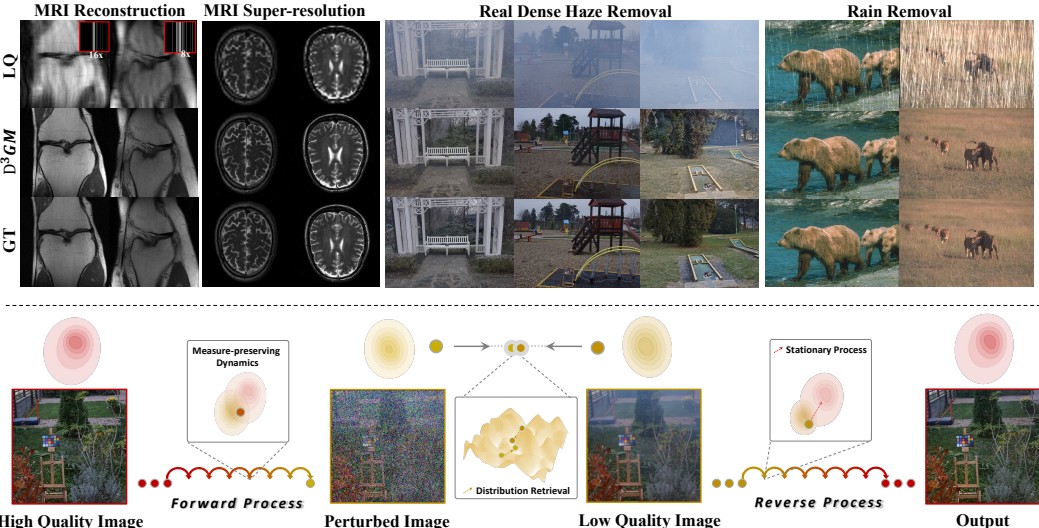

Figure 1: Dynamics-aware SDE Diffusion Generative Model (D$^3$GM). When extending transitionary SDEs to random dynamical systems (RDS), their measure-preserving property should be kept to maintain stability. This corresponds to driving the SDE towards the drift term $\mu$ (LQ). There is a *Temporal Distribution Discrepancy* which results from the gap between the forward estimation $x_T$ and the low quality image in the SDE. With the distribution aligned between $x_T$ and $\mu$, the SDE can be made more robust to inverse problems. Reconstruction results for low quality (LQ) images after application of our D$^3$GM method, on different tasks, compared to the ground truth (GT) on two domains - The frequency domain: MRI Reconstruction (undersampling factor 8x, 16x, frequency masks are colored red); MRI Super-resolution (up-scaling factor of X4, cross-domain evaluation). The image domain: Real Dense Haze Removal; Rain Removal (light, heavy).

> **Example 2.** When $\frac{\sigma}{\theta} \to \infty$, the attractor becomes excessively large, reducing the significance of $\mu$. SDEs exhibiting this behavior are defined as Coefficient Decoupled SDEs. In practice, $\mu$ demonstrates non-infinite properties as an input image, while the corresponding sigma and theta are indeed unconstrained. In such decoupled forms, the coefficients of the attractor size increases $\frac{\sigma}{\mu}$, diminishing the significance of $\mu$.

Based on Prop. 2, since the temporal distribution discrepancy always exists as long as the running time $T$ is finite, and we want the final state $x_T$ to be as close as possible to the distribution of $x_\infty$. Therefore, we introduce $\tau$ such that given $T$, the distribution of $x_T$ follows $\mathcal{N}(\mu(1 - e^{\bar{\theta}_T}) + x_0 e^{\bar{\theta}_T}, \tau^2 \lambda^2 (1 - e^{2\bar{\theta}_T}))$, and $x_\infty$ follows $\mathcal{N}(\mu, \tau^2\lambda^2)$, with $\tau > 1$, we increase the possibility of a sample $\tilde{x}_T$ from $x_T$ to become closer to the distribution of $x_\infty$, and thus serves as a plausible initial state for the reverse process. We can control how much to close the distributions, either by increasing the stiffness $\tau$ at the cost of potentially destabilizing the reverse process, or by decreasing $\tau$ to further smooth the density functions of both distributions at the cost of more reverse iterations.

By connecting the inverse problem with the analysis above, we clarify the discrepancy in the stationary modeling process from measure-preserving dynamics and thereby improve the generalization of diffusion processes and the accuracy of the reverse process. This is particularly important for accommodating the diversity of degradation states and to ensure accurate sampling.

**Forward Process.** We describe the forward process as: $dx_t = \theta_t(\mu - x_t)dt + \tau\sigma_t dW_t$, parameterized by $\tau$ to calibrate the SDE modeling, $\mu$ is the state mean. The parameters $\theta_t$ and $\sigma_t$, both being time-dependent and strictly positive, correspond to the rate of mean reversion and the stochastic volatility, respectively. The selection of $\theta_t$ and $\sigma_t$ offers flexibility in Tab. 2, *c.p.*, Sect. 3.

Considering the trade-off between complexity and effectiveness, Cos has been chosen for both $\theta_t$ and $\sigma_t$. This aims at capturing complex temporal dynamics in a computationally tractable manner, thereby optimizing the balance between the performance and calculation convenience.

Table 2: $\mu_t(x_t, t)$, $v_t(x_t, t)$ solutions with various $\theta_t$, $\sigma_t$.

| | $\mu_t(x_t, t)$ | $v_t(x_t, t)$ |
|---|---|---|
| Lin | $e^{\theta \frac{x^2}{2}} \mathbf{x}_0 + \left(1 - e^{\theta \frac{x^2}{2}}\right) \mathbf{y}$ | $\tau^2 \lambda^2 \left(1 - e^{\theta x^2}\right) \mathbf{y}$ |
| Log | $e^{\frac{\theta}{k} \log(1+e^{kt})} \mathbf{x}_0 + \left(1 - e^{\frac{\theta}{k} \log(1+e^{kt})}\right) \mathbf{y}$ | $\tau^2 \lambda^2 \left(1 - e^{\frac{2\theta}{k} \log(1+e^{kt})}\right) \mathbf{y}$ |
| Cos | $e^{-\theta\left(t - \frac{\sin(\theta t)}{\theta}\right)} \mathbf{x}_0 + \left(1 - e^{-\theta\left(t - \frac{\sin(\theta t)}{\theta}\right)}\right) \mathbf{y}$ | $\tau^2 \lambda^2 \left(1 - e^{-2\theta\left(t - \frac{\sin(\theta t)}{\theta}\right)}\right) \mathbf{y}$ |
| Quad | $e^{\theta \frac{x^3}{3}} \mathbf{x}_0 + (1 - e^{\theta \frac{x^3}{3}}) \mathbf{y}$ | $\tau^2 \lambda^2 (1 - e^{2\theta \frac{x^3}{3}}) \mathbf{y}$ |

In the forward process, the mean $\mu_t$ approaches the low-quality image with $\mathrm{E}(x_t) = \mu$, while the variance tends toward the stationary variance $\mathrm{var}(x_t) = \frac{\tau^2 \sigma^2}{2\theta}$. Essentially, the forward SDE transitions the high-quality image to a low-quality counterpart infused with Gaussian noise. The discretized SDE for the forward process is $x_{t_i} = x_{t_{i-1}} + \theta_{t_{i-1}}(\mu - x_{t_{i-1}})\Delta t + \tau \sigma_{t_{i-1}} \Delta W_i$. We employ a transition strategy utilizing a varied stationary variance. Additionally, we execute an unconditional update, which operates without the need for matching in the reverse process. These not only allow image corruption but also provides effective adaptability for improvements.

**Reverse Process.** The reverse process aims at reconstructing the original image by gradually denoising a low quality image. It utilizes the score of the marginal distribution, denoted as $\nabla_x \log \hat{p}_t(x)$, and is governed by:

$$\mathrm{d}x_t = \left[\theta_t(\mu - x_t) - \tau^2 \sigma_t^2 \nabla_x \log \hat{p}_t(x)\right] \mathrm{d}t + \tau \sigma_t \mathrm{d}\widehat{W}_t. \tag{6}$$

The reverse-time D$^3$GM process of Eq. 6 can be found in Appx. D. This closely mirrors the forward process and incorporates an additional drift term proportional to the score of the marginal distribution. The ground truth score for this process, necessary for training our generative model, is:

$$\nabla_x \log \hat{p}_t(x \mid x_0) = -\frac{x_t - \mu_t(x)}{v_t}, \tag{7}$$

where $\mu_t(x)$ represents the random attractor of the process at time $t$, and $v_t$ is the variance. Our training objective is defined as the minimization of the expected discrepancy between the predicted and true scores over the data distribution:

$$\theta^* = \arg\min_\theta \mathbb{E}_{t,(\mathbf{x}_0,\mathbf{y}),\mathbf{z},\mathbf{x}_t} \left[w \left\| S_\theta(\mathbf{x}_t, \mathbf{y}, t) - \nabla_{\mathbf{x}_t} \log p_{0t}(\mathbf{x}_t \mid \mathbf{x}_0, \mathbf{y})\right\|_2^2\right], \tag{8}$$

where $w = -1/\tau^2$ is a time-dependent weighting function, and $S_\theta$ denotes the score network parameterized by $\theta$ which approximates the score of the marginal distribution. The optimization is conducted over the network parameters $\theta$, under the expectation with respect to the time variable $t$, the initial image $\mathbf{x}_0$, noisy image $\mathbf{x}_t$, and data $\mathbf{y}$.

## 5 Experiments

**Experimental Settings:** We evaluate D$^3$GM on various challenging restoration and reconstruction problems. We initially analyze our method by examining its performance with closely related diffusion formulation variants. Subsequently, we benchmark D$^3$GM against the state-of-the-art techniques in these domains. For comprehensive evaluation across all experiments, we report the PSNR [25] and SSIM [58] for pixel- and structural-level alignment, LPIPS [72] for measuring perceptual variance. An in-depth description of our implementation is provided in Appx. G.

### 5.1 Stability: Illustrations of the Measure-preserving Dynamics within Diffusion Models

**SGM, and Transitionary SGMs *vs.* D$^3$GM:**

We perform qualitative and quantitative analyses using variants of closely related formulations for Prop. 1 and 2 and evaluate across (A) SGMs and (B) transitionary SGMs. (A) uses a common score-based SDE, (B) uses a Coefficient Decoupled SDE (*e.g.*, variance exploding SDE with the drift term $\mu$) according to Prop. 1 and OU SDE, alongside our D$^3$GM.

Table 3: Quantitative results for Rain100H and Rain100L.(best in bold and second best underlined)

| Method | Rain100H | | | Rain100L | | |
|---|---|---|---|---|---|---|
| | PSNR↑ | SSIM↑ | LPIPS↓ | PSNR↑ | SSIM↑ | LPIPS↓ |
| JORDER [64] | 26.25 | 0.835 | 0.197 | 36.61 | 0.974 | 0.028 |
| IRCNN [37] | 29.12 | 0.882 | 0.153 | 33.17 | 0.958 | 0.068 |
| PreNet [49] | 29.46 | 0.899 | 0.128 | 37.48 | 0.979 | 0.020 |
| MPRNet [68] | 30.41 | 0.891 | 0.158 | 36.40 | 0.965 | 0.077 |
| MAXIM [56] | 30.81 | 0.903 | 0.133 | 38.06 | 0.977 | 0.048 |
| VPB (CD) [73] | 30.89 | 0.885 | 0.051 | 38.12 | 0.968 | 0.023 |
| IR-SDE (OU) [37] | 31.65 | 0.904 | 0.047 | 38.30 | 0.981 | 0.014 |
| $D^3GM$ | **32.41** | **0.912** | **0.040** | **38.40** | **0.982** | **0.013** |

Table 4: Quantitative results for O-HAZE and Dense-Haze.

| Methods | O-HAZE | | Dense-Haze | |
|---|---|---|---|---|
| | PSNR↑ | SSIM↑ | PSNR↑ | SSIM↑ |
| DCP [23] | 16.78 | 0.653 | 12.72 | 0.442 |
| DehazeNet [6] | 17.57 | 0.770 | 13.84 | 0.430 |
| GFN [50] | 18.16 | 0.671 | - | - |
| GDN [36] | 18.92 | 0.672 | 14.96 | 0.536 |
| MSBDN [18] | 24.36 | 0.749 | 15.13 | 0.555 |
| FFA-Net [46] | 22.12 | 0.770 | 15.70 | 0.549 |
| AECR-Net [60] | - | - | 15.80 | 0.466 |
| SGID-PFF [4] | 20.96 | 0.741 | 12.49 | 0.517 |
| Restormer [67] | 23.58 | 0.768 | 15.78 | 0.548 |
| Dehamer [22] | 25.11 | 0.777 | **16.62** | 0.560 |
| MB-TF [47] | 25.31 | 0.782 | 16.44 | **0.566** |
| $D^3GM$ | **26.23** | **0.786** | 15.85 | 0.551 |

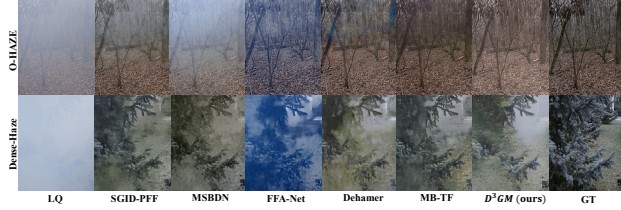
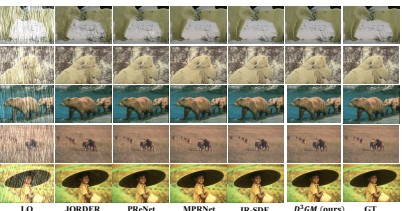

(a) O-HAZE and Dense-Haze.   (b) Rain100H and Rain100L.

Figure 3: Qualitative results for (a) deraining and (b) dehazing.

Following Tab. 1, VPB [73] can be regarded as a Coefficient Decoupled SDE, and IR-SDE [37] as an OU SDE. Our results in Fig. 2 illustrate that $D^3GM$ converges stably towards the expected distribution, unlike other methods which exhibit instability or deviation. This highlights the reliance of other techniques, *e.g.*, score-based SDEs, on retrospective measurement consistency corrections.

**Simulated Deraining:** We evaluated $D^3GM$ together with the state-of-the-art deraining strategies: (1) OU SDE method IR-SDE [37], Coefficient Decoupled (CD) VPB [73] and other CNNs [64, 49, 68, 56]. We use two of the most renowned synthetic raining datasets: Rain100H [65] and Rain100L [65]. Rain100H contains 1800 pairs of images with and without rain, along with 100 test pairs. As for Rain100L,

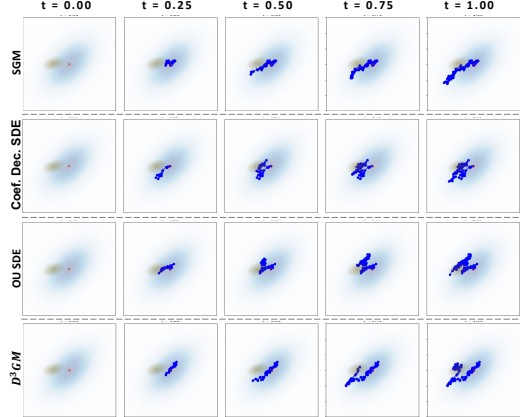

Figure 2: Sampling trajectories of SGM, transitionary SGMs: Coef. Dec., OU SDE, and $D^3GM$.

it consists of 200 pairs for training and 100 pairs for testing. We present results based on the PSNR, SSIM, and LPIPS metrics.

Quantitative results from the two raining datasets are presented in Tab. 3. Based on both distortion and perceptual metrics, $D^3GM$ is capable of generating the most realistic and high fidelity results as shown in Fig. 3b.

## 5.2 Generalizability: $D^3GM$ for real-world data

**Case Study 1: Dehazing.** We utilize the real-world datasets O-HAZE [2], Dense-Haze [1], which contain 45 and 55 paired images, respectively. We use the last 5 images of each dataset as the testing set and the rest as the training set following the common split of other methods. Results are shown in Tab. 4 and Fig. 3a. Our work improves results on O-HAZE both quantitatively and qualitatively. Smaller improvements are observed on Dense-Haze. This can be attributed to the severe signal corruption of the Dense-Haze data. A combination with tailored task-specific, Transformer-based

methods [47, 22] might lead to further performance gains for such data. Such extensions are beyond the focus of this paper. Qualitatively D$^3$GM achieves excellent visual results (Fig. 1 and Fig. 3a).

**Case Study 2: MRI Reconstruction.** MRI data is represented in the complex-valued frequency domain, which is distinctly different from the natural image domain. We utilized the fastMRI dataset [69], containing single-channel, complex-valued MRI samples. Implementation details can be found in Appx. G. For a robust comparison, we benchmarked against a diverse set of deep learning-based state-of-the-art reconstruction methods. Although our method does not have a task-specific design, we still get comparable performance (more details and results are provided in Appx. H). Fig. 1 illustrates our reconstruction results from masked k-space data for 8x and 16x acceleration, *i.e.*, under-sampling for faster data acquisition.

Table 5: Quantitative results for fastMRI dataset with acceleration rates x8 and x16.

| Method | Undersampling factor 8x | | | Undersampling factor 16x | | |
|---|---|---|---|---|---|---|
| | PSNR↑ | SSIM↑ | LPIPS↓ | PSNR↑ | SSIM↑ | LPIPS↓ |
| ZeroFilling | 22.74 | 0.678 | 0.504 | 20.04 | 0.624 | 0.580 |
| D5C5 [51] | 25.99 | 0.719 | 0.291 | 23.35 | 0.667 | 0.412 |
| DAGAN [62] | 25.19 | 0.709 | 0.262 | 23.87 | 0.673 | 0.317 |
| SwinMR [27] | 26.98 | 0.730 | 0.254 | 24.85 | 0.673 | 0.327 |
| DiffuseRecon [44] | 27.40 | 0.738 | 0.286 | 25.75 | 0.688 | 0.362 |
| CDiffMR [26] | 27.26 | **0.744** | 0.236 | **25.77** | **0.707** | 0.293 |
| D$^3$GM | **27.92** | 0.740 | **0.175** | 25.26 | 0.701 | **0.153** |

Table 6: Quantitative results for IXI MRI SR on unseen datasets.

| | HH | Guys | | IOP | |
|---|---|---|---|---|---|
| | Methods | PSNR↑ | SSIM↑ | PSNR↑ | SSIM↑ |
| (X4, IXI T2w) | EDSR [33] | 23.03 | 0.700 | 25.10 | 0.727 |
| | SFM [19] | 23.28 | 0.711 | 25.18 | 0.731 |
| | PDM [70] | 22.89 | 0.709 | 27.93 | 0.851 |
| | ACT [48] | 22.80 | 0.707 | 26.38 | 0.826 |
| | CST [14] | 23.70 | 0.714 | 28.55 | 0.837 |
| | D$^3$GM | 25.13 | 0.799 | 28.60 | 0.863 |

**Case Study 3: MRI Super-resolution (SR).** IXI[3] dataset is the largest benchmark considered in our MRI SR evaluation. Clinical MRI T2-weighted (T2w) scans are collected from three hospitals with different imaging protocols: HH, Guys, and IOP. For investigating our cross-domain generalization and robustness, a challenging task for both MRI SR and natural image restoration, we trained on HH data with k-space truncation, and tested on Guys and IOP with kernel degradation with an up-scaling factor of X4. More details can be found in Appx. G.

The methods are tested under unseen data conditions, including different acquisition parameters, MRI scanners (different vendors and field-strengths) and unseen degradations. With D$^3$GM we are able to demonstrate varying degrees of improvement, as well as generalizability to the discrepancy within the training domain and across the test domain as shown in Tab. 6. Qualitative results are shown in Fig. 11 and further results across domains in Appx. H.

## 6 Discussion and limitations

Other works, like VPB (CD) [73], I$^2$SB [34] are based on diffusion bridges assuming that clean and degraded images are already close. Thus, the tractability of the reverse process heavily relies on the validity of the assumed Dirac delta distribution. IR-SDE [37] employs the mean-reverting SDE theorem based on running the reverse SDE with *instability*. Since unstable errors accumulate in each step, this model will eventually become unable to learn the transformation, *e.g.*, degradation. DPS [9] and CDDB [10] assume that the degradation process is known, or linear operations are directly used to simulate the degradation process, which limits the generalizability of the method. In contrast, D$^3$GM is built on the theorem of Measure-preserving RDS, which bridges clean and degraded image distributions while taking both degradation and measurements into account. Moreover, D$^3$GM can be extended to a two-sided solution operator (tractability) with a flow map according to Prop. 1.

**Limitations.** Even though our results are better than others when the degradation process is very severely corrupted (*e.g.,* real dehazing), the overall quality of the restored image is still limited, which is consistent with Prop. 2. This might be alleviated via guiding the sampling process with priors and enhanced $\mu$, such as posterior sampling or degradation maps on the data manifold, but such approaches are still limited as shown in Tab. 7 and Appx I.

**Computational Complexity *vs.* Performance.** Tab. 7 highlights that prior work is often tailor-made for a specific subset of tasks and thus also generalisation-limited for challenging environments in practice. D$^3$GM's focus on a generic robust solutions from an RDS perspective can mitigate this, while maintaining en-par performance with *task-specific* approaches in Tab. 8.

---

[3]http://brain-development.org/ixi-dataset/

Table 7: Comparison to tailor-made approaches.

| Real Dehazing | Diffu. for Inverse Problems. | | Transitionary SGMs. | | |
|---|---|---|---|---|---|
| **Methods** | DPS [9] | CDDB [10] | $I^2SB$ [34] | IR-SDE [37] | $D^3GM$ |
| **PSNR↑** | 18.63 | 21.55 | 21.51 | 24.52 | **26.23** |
| **SSIM↑** | 0.448 | 0.591 | 0.583 | 0.691 | **0.786** |

Table 8: $D^3GM$ vs. recent deraining works.

| Deraining | rain200H | | Model Complexity | |
|---|---|---|---|---|
| **Methods** | **PSNR↑** | **SSIM↑** | **Param.** | **FLOPs** |
| DRSformer [7] | 32.17 | 0.933 | 33.7M | 242.9G |
| $D^3GM$ | **32.21** | 0.925 | 36.5M | **104.7G** |

# 7 Conclusion

The proposed $D^3GM$ framework enhances the stability and generalizability of SDE-based diffusion methods for challenging inverse problems. Our approach, grounded in measure-preserving dynamics of random dynamical systems, ensures broad applicability and relevance. We demonstrate $D^3GM$'s effectiveness across various benchmarks, including challenging tasks like MRI reconstruction.

**Acknowledgements:** This work was supported by the JADS programme and UK Research and Innovation [UKRI Centre for Doctoral Training in AI for Healthcare grant number EP/S023283/1]. HPC resources were provided by the Erlangen National High Performance Computing Center (NHR@FAU) of the Friedrich-Alexander-Universität Erlangen-Nürnberg (FAU) under the NHR project b143dc and b180dc. NHR funding is provided by federal and Bavarian state authorities. NHR@FAU hardware is partially funded by the German Research Foundation (DFG) – 440719683. Support was also received by the ERC - projects MIA-NORMAL 101083647 as well as DFG 513220538, 512819079.

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

# Appendix

## A  Broader Impacts

Our proposed method offers significant advancements in the restoration of degraded images with minimal risk of hallucinations due to stability guarantees and with applications including MRI reconstruction and super-resolution. In the clinical domain, the adoption of our method for MRI reconstruction must adhere to stringent regulatory and approval processes. The results generated by our model should serve as an auxiliary tool to assist healthcare professionals in their diagnostic and treatment decisions, rather than as a standalone diagnostic tool.

## B  Intuition of Measure-Preserving Dynamics in SDE:

Consider the analogy of a stretched rubber band, which naturally seeks to return to its original but does so with a lot of oscillations when released. This elastic behavior parallels the dynamics of the OU process, where deviations from a mean state are counteracted by a restorative force, guiding the system back towards equilibrium (*i.e.*, final state), with random perturbations.

Our process models the noise as a stochastic component that fluctuates around a stationary process and improve the OU process with RDS. Measure-preserving dynamics ensure that while the image undergoes transformations during the denoising process, the overall statistical properties remain consistent (*i.e.*, invariant image features), which cannot be satisfied by vanilla OU processes or previous approaches (Tab. 1).

## C  Mathematical Preliminaries

Consider a probability space $(\Omega, \mathcal{F}, P)$ accompanied by a standard Brownian motion $W_t$. A stochastic process $x_t$ over the interval $0 \leq t \leq T$ can be formulated by the following stochastic differential equation (SDE):

$$dx_t = b(t, x_t)dt + \sigma(t, x_t)dW_t \tag{9}$$

**Definition 1** *Filtration A collection of sigma-fields,*

$$\mathbb{F} := \{\mathcal{F}_t, 0 \leq t \leq T\},$$

*is termed a filtration if:*

1. *$\mathcal{F}_t \subset \mathcal{F}$ is a sub-$\sigma$-field for every $t \in [0, T]$;*

2. *If $0 \leq t_1 < t_2 \leq T$, then $\mathcal{F}_{t_1} \subset \mathcal{F}_{t_2}$.*

*Here, $\mathcal{F}_t$ represents the information set at time t.*

**Definition 2** *Strong Solution A process $x$, which is $\mathcal{F}$-progressively measurable, is considered a strong solution to the SDE given by Eq. 1 if:*

$$\int_0^T (|b(t, x_t)|^2 + |\sigma(t, x_t)|^2)dt < \infty$$

*almost surely. This is captured by:*

$$x_t = x_0 + \int_0^t b(s, x_s)ds + \int_0^t \sigma(s, x_s)dW_s, \forall t \in [0, T] \tag{10}$$

**Definition 3** *Lipschitz Continuity For an $N$-dimensional stochastic process $x_t$ over $t \in [0, \infty)$, adapted to the filtration $\mathcal{F}$, function $w(x_t)$ exhibits Lipschitz continuity in $x$ if:*

1. *$w(x_t)$ is $\mathcal{F}$-measurable with the requisite dimensions.*

2. *There exists a non-negative constant $K$ such that, for every $x_t$ and $x_s$,*

$$|w(x_t) - w(x_s)| \leq K|x_t - x_s|.$$

*In numerous diffusion models, Lipschitz continuity is inherent, ensuring the existence and uniqueness of the solution to the stochastic process. However, for clarity in network design, the emphasis on Lipschitz continuity ensures the foundation of neural networks remains consistent.*

**Definition 4** *Random Dynamical System A random dynamical system(RDS) consists of a base flow, the "noise", and a cocycle dynamical system on the "physical" phase space, we first discuss one fundamental element of our RDS, the base flow.*
*Let $(\Omega, \mathcal{F}, \mathbb{P})$ be a probability space, the noise space. Define the base flow $\vartheta : \mathbb{R} \times \Omega \to \Omega$ as follows: for each "time" $s \in \mathbb{R}$, let $\vartheta_s : \Omega \to \Omega$ be a measure-preserving measurable function: $\mathbb{P}(E) = \mathbb{P}\left(\vartheta_s^{-1}(E)\right)$ for all $E \in \mathcal{F}$ and $s \in \mathbb{R}$*
*Suppose also that*
*1. $\vartheta_0 = \mathrm{id}_\Omega : \Omega \to \Omega$, the identity function on $\Omega$;*
*2. for all $s, t \in \mathbb{R}, \vartheta_s \circ \vartheta_t = \vartheta_{s+t}$.*
*That is, $\vartheta_s, s \in \mathbb{R}$, forms a group of measure-preserving transformation of the noise $(\Omega, \mathcal{F}, \mathbb{P})$*
*Now we are ready to define the **random dynamical system(RDS)**.*
*let $(X, d)$ be a complete separable metric space, the phase space. Let $\varphi : \mathbb{R} \times \Omega \times X \to X$ be a $(\mathcal{B}(\mathbb{R}) \otimes \mathcal{F} \otimes \mathcal{B}(X), \mathcal{B}(X))$ measurable function such that 1. for all $\omega \in \Omega, \varphi(0, \omega) = \mathrm{id}_X : X \to X$, the identity function on $X$; 2. for (almost) all $\omega \in \Omega, (t, x) \mapsto \varphi(t, \omega, x)$ is continuous; 3. $\varphi$ satisfies the (crude) cocycle property: for almost all $\omega \in \Omega$,*

$$\varphi\left(t, \vartheta_s(\omega)\right) \circ \varphi(s, \omega) = \varphi(t + s, \omega)$$

*In the case of random dynamical systems driven by a Wiener process $W : \mathbb{R} \times \Omega \to X$, the base flow $\vartheta_s : \Omega \to \Omega$ would be given by*

$$W\left(t, \vartheta_s(\omega)\right) = W(t + s, \omega) - W(s, \omega).$$

**Theorem 1** *Existence and Uniqueness If the initial condition $x_0 \in \mathbb{L}^2$ is a random variable that's independent of $W$ and both $\mu(0, x_0)$ and $\sigma(0, x_0) \in \mathbb{H}^2$, then, provided there exists a constant $K > 0$ that satisfies:*

$$
\begin{aligned}
&|b(t, x) - b(t, y)| + |\sigma(t, x) - \sigma(t, y)| \\
&\leq K|x - y|, \quad \forall t \in [0, T], x, y \in \mathbb{R}^n
\end{aligned}
\tag{11}
$$

*(an attribute also recognized as Lipschitz continuity), a unique strong solution to Eq. 1 exists in $\mathbb{H}^2$ for every $T > 0$. Additionally:*

$$\mathbb{E}\left[\sup_{t \leq T} |x_t|^2\right] \leq C(1 + \mathbb{E}|x_0|^2)e^{CT} \tag{12}$$

*holds true, where the constant $C$ depends on both $T$ and $K$.*

# D    Preliminaries and Proof for Proposition 1

## D.1    Proof for reverse-time D³GM process:

There is a one-to-one and onto correspondence between the stochastic differential equation and the Kolmogorov equation for $p\left(x_t, t \mid x_s, s\right), t \geqslant s$, which describes the evolution of the underlying probability distribution. Consequently, there should be a one-to-one and onto correspondence between a reverse-time equation for $\tilde{x}_t$ and a Kolmogorov equation for $p(x_t, t | x_s, s), s \geqslant t$

$$dx_t = \theta_t(\mu - x_t)dt + \tau\sigma dW_t$$

We have the corresponding Kolmogorov backward equation given by

$$
\begin{aligned}
-\frac{\partial p\left(x_s, s \mid x_t, t\right)}{\partial t} =& \theta_t(\mu - x_t) \cdot \frac{\partial p\left(x_s, s \mid x_t, t\right)}{\partial x_t} \\
&+ \frac{1}{2}\tau^2\sigma_t^2 \cdot \frac{\partial^2 p\left(x_s, s \mid x_t, t\right)}{\partial x_t^2},
\end{aligned}
\tag{13}
$$

The unconditioned Kolmogorov forward equation is given by

$$
\begin{aligned}
-\frac{\partial p\left(x_s, s \mid x_t, t\right)}{\partial t} = & \frac{\partial\left(\theta_t(\mu - x_t) \cdot p\left(x_s, s \mid x_t, t\right)\right)}{\partial x_t} \\
& -\frac{1}{2} \cdot \frac{\partial^2\left(\tau^2 \sigma_t^2 \cdot p\left(x_s, s \mid x_t, t\right)\right)}{\partial x_t^2},
\end{aligned}
\tag{14}
$$

See [3] for more details on Kolmogorov equations.
Bayes rule gives

$$
p\left(x_t, t, x_s, s\right) = p\left(x_s, s \mid x_t, t\right) p\left(x_t, t\right)
$$

We plug this result into 13, which gives us the Kolmogorov equation

$$
\begin{aligned}
-\frac{\partial}{\partial t} p\left(x_t, t, x_s, s\right) = & \frac{\partial}{\partial x_t}\left[\bar{f}\left(x_t, t\right) p\left(x_t, t, x_s, s\right)\right] \\
& +\frac{1}{2} \frac{\partial^2\left[p\left(x_t, t, x_s, s\right) \cdot \tau^2 \sigma_t^2\right]}{\partial x_t^2}
\end{aligned}
\tag{15}
$$

and the expression for $\bar{f}$ is given by

$$
\begin{aligned}
\bar{f}\left(x_t, t\right) & = \theta_t(\mu - x_t) - \frac{1}{p\left(x_t, t\right)} \frac{\partial}{\partial x_t}\left[p\left(x_t, t\right) \tau^2 \sigma_t^2\right] \\
& = \theta_t(\mu - x_t) - \tau^2 \sigma_t^2 \log \frac{\partial}{\partial x_t}\left[p(x_t, t)\right]
\end{aligned}
\tag{16}
$$

Therefore, we have that the reverse process corresponds to the Kolmogorov equation 16 is given by

$$
dx_t = \theta_t(\mu - x_t) - \tau^2 \sigma_t^2 \log \nabla_x p_t(x_t) + \tau^2 \sigma_t^2 d\bar{W}_t
$$

**Definition 5** *Different from deterministic dynamical systems, random dynamical systems usually consider a pullback attractor rather than a forward attractor due to the non-autonomousness introduced by the random noise. The pullback attractor (or random global attractor) $\mathcal{A}(\omega)$ for the RDS $\varphi$ we defined in 1 is a $\mathbb{P}$-almost surely unique random set such that:*
*1. $\mathcal{A}(\omega)$ is a random compact set: $\mathcal{A}(\omega) \subseteq X$ is almost surely compact and $\omega \mapsto \mathrm{d}(x, \mathcal{A}(\omega))$ is a $(\mathcal{F}, \mathcal{B}(X))$-measurable function for every $x \in X$*
*2. $\mathcal{A}(\omega)$ is invariant: for all $\varphi(t, \omega)(\mathcal{A}(\omega)) = \mathcal{A}(\vartheta_t\omega)$ almost surely;*
*3. $\mathcal{A}(\omega)$ is attractive: for any deterministic bounded set $B \subseteq X$, $\lim_{t \to +\infty} \mathrm{d}\left(\varphi\left(t, \vartheta_{-t}\omega\right)(B), \mathcal{A}(\omega)\right) = 0$ almost surely.*
*$\mathcal{B}(X)$ denotes the Borel $\sigma$-algebra generated by the space $X$ where the RDS is defined.*

**Definition 6** *Poincare Recurrence Theorem*
*Let*

$$
(X, \Sigma, \mu)
$$

*be a finite measure space and let*

$$
f : X \to X
$$

*be a measure-preserving transformation. then we have that for any $E \in \Sigma$, the set of those points $x$ of $E$ for which there exists $N \in \mathbb{N}$ such that $f^n(x) \notin E$ for all $n > N$ has zero measure.*
*In other words, almost every point of E returns to E. In fact, almost every point returns infinitely often; i.e. $\mu\left(\{x \in E : \text{ there exists } N \text{ such that } f^n(x) \notin E \text{ for all } n > N\}\right) = 0$.*

### D.2 Analysis of Proposition 1

**Proposition 1** After extending the solution of the OU process to RDS, the measure-preserving flow map of the solution should meet the property $\varphi(t, s; \omega)x = \varphi\left(t - s, 0; \vartheta_s\omega\right)x$. However, OU processes with time-varying coefficients are usually not satisfied for this property(can be referred to as time-homogeneity) and thus the stability of the system breaks.
The forward process in SDE notation

$$
dx_t = \theta_t(\mu - x_t)dt + \sigma_t dW_t
$$

and the solution to the above SDE is given by

$$x_t = \mu + (x_s - \mu)e^{-\bar{\theta}_{s:t}} + \int_s^t \sigma_z e^{-\bar{\theta}_{z:t}} dW_z$$

where $\bar{\theta}_{s:t} = \int_s^t \theta_z dz$.

This solution above can be represented by a continuous random dynamical system (RDS) $\varphi$ defined on a complete separable metric space $(X, d)$, where the noise is chosen from a probability space $(\Omega, \mathcal{F}, \mathbb{P})$. More details can be found in [13].

More generally, we can extend the RDS to two-sided, infinite time, define a flow map or (solution operator) $\varphi : \mathbb{R} \times \Omega \times \mathbb{R}^d \to \mathbb{R}^d$ by $\varphi(t, s, |\omega, x_0) := x(t, s|\omega, x_0)$ with $\omega \in \Omega$, $-\infty < s \leqslant t < \infty$. The base flow driven by Brownian motion can be explicitly written as $W(t, \vartheta_s(\omega)) = W(t+s, \omega) - W(s, \omega)$.

Now suppose that:

1. The flow map $\vartheta_t, t \in \mathbb{R}$ is a measure-preserving transformations of $(\Omega, \mathcal{F}, P)$, with the property that for all $s < t$ and $x \in X$,

$$\varphi(t, s; \omega)x = \varphi(t - s, 0; \vartheta_s\omega)x, \quad P\text{-a.s.} \tag{17}$$

2. (i) $\varphi(t, r; \omega)\varphi(r, s; \omega)x = \varphi(t, s; \omega)x$ for all $s \leqslant r \leqslant t$ and $x \in X$;

(ii) $\varphi(t, s; \omega)$ is continuous in $X$, for all $s \leqslant t$.

(iii) for all $s < t$ and $x \in X$, the mapping

$$\omega \mapsto \varphi(t, s; \omega)x$$

is measurable from $(\Omega, \mathcal{F})$ to $(X, \mathcal{B}(X))$; and

(iv) for all $t, x \in X$, and $P$-a.e. $\omega$, the mapping $s \mapsto \varphi(t, s; \omega)x$ is right continuous at any point. Where $\mathcal{B}(X)$ denotes the $\sigma$-algebra generated by X.

Under assumptions (i), (ii), (iii), (iv) and suppose that for $P$-a.e. $\omega$ there exists a compact attracting set $K(\omega)$ at time 0, *i.e.*, such that for all bounded sets $B \subset X$,

$$d(\varphi(0, s; \omega)B, K(\omega)) \to 0 \quad \text{as} \quad s \to -\infty$$

We can see that the attractor of this system is defined in the pullback sense, such that time is rewind backward before iterating forward.

Moreover, the reverse process with any starting time $t$ to $s$ is defined as the RDS going backward in time

$$\varphi(s, t | \vartheta_t\omega, x_t)$$

start from the time $t$ realization and run backwards to $s$.

the above proposition can be extended to show that there exists a compact attracting set at any $-\infty < t < \infty$. and this convention has allowed us to characterize the attractor $K(\omega) = \mathcal{N}(\mu, \lambda^2)$, when the time becomes finite, for example from 0 to $T$, the random attractors can be abstractly viewed as $\mathcal{N}\left(\mu + (x_s - \mu)e^{-\bar{\theta}_{0:t}}, \lambda(1 - e^{-\bar{\theta}_{0:t}})\right)$.

Moreover, an important assumption in the 17 is usually not satisfied for OU processes with time-varying coefficients, therefore, we impose that for every $t$, $\frac{\sigma_t^2}{2\theta_t} = \lambda^2$, where $\lambda$ is a constant, and will be the asymptotic variance of the forward process. This convention has allowed us to reduce the regularization on two variables $\sigma_t, \theta_t$ to just one variable to satisfy 17; and this convention has allowed as to characterize the attractor $K(\omega) = \mathcal{N}(\mu, \lambda^2)$, when the time becomes finite, for example from 0 to $T$, the random attractors can be abstractly viewed as the Gaussian measure $\mathcal{N}\left(\mu + (x_s - \mu)e^{-\bar{\theta}_{0:t}}, \lambda(1 - e^{-\bar{\theta}_{0:t}})\right)$.

## E   Proof for Proposition 2

**Proposition 2** Given Eq. 3 and Eq. 4, and assume that the score function is bounded by $C$ in $L^2$ norm, then the discrepancy between the reference and the retrieved data is, with probability at least $(1 - \delta)$:

$$\|x_0 - \mathrm{OU}(x_0, \mu; T, \theta)\|_2^2$$
$$\geq | \left( (x_0 - \mu)^2 - \sigma_T^2/2\theta_T \right) \mathrm{e}^{-2\bar{\theta}_T} + \sigma_T^2/2\theta_T \tag{18}$$
$$- \sigma_{max}^2 \left( C\sigma_{max}^2 + d + 2\sqrt{-d \cdot \log \delta} - 2\log \delta \right) |,$$

where $x_0$ $\hat{x}_0$ are the quality reference and sampling data. For a noisy inverse problem scenario, the retrieved data with any finite $T$ always indicates difference depends on $\sigma_t, \mu_t, \lambda, T, \bar{K}$, where $\bar{K}$ is the Lipschitz constant for the reverse process.

we have that the absolute value between the theoretical expectation and actual expectation after $T$ period is given by

$$\|\mu - \mathbb{E}(\hat{x}_T)\| = \|(x_0 - \mu)e^{-\bar{\theta}_T}\| > 0$$

Similarly, the difference between theoretical variance and T-period variance also has a strictly positive difference. where $\bar{\theta}_t = \int_0^t \theta_s ds$.

Therefore, with finite $T$, the final state of the forward process can only reach a $\hat{x}_T$ rather than the theoretical stationary distribution, which we denote by $x_\infty$, we denote the retrieved image after T-periods from theoretical stationary distribution by $\hat{x}_0$, $x_0$ by the ground truth HQ image, $\hat{x}_T$ the true distribution after T iteration,

$$\|x_Q - f_{\mathrm{OU}}n(x_Q, \mu; t_0,)\| = \|\hat{x}_T - x_\infty - (\hat{x}_0) - x_\infty\|_2^2$$
$$\geq \|\|\hat{x}_T - x_\infty\|_2^2 - \|\hat{x}_0 - x_\infty\|_2^2\|_2^2 \tag{19}$$

Inside the norm, the first term is bounded below, since both $\hat{x}_T$ and $x_\infty$ both follow a normal distribution and are independent of each other, the difference between those two random variables, we denote by $z_T$, that follows a normal distribution

$$\mathcal{N}\left(\mu + (x_0 - \mu)\mathrm{e}^{-\bar{\theta}_t} - \mu, \lambda^2(1 - e^{-2\bar{\theta}_t}) + \lambda^2\right)$$
$$= \mathcal{N}\left((x_0 - \mu)\mathrm{e}^{-\bar{\theta}_t}, \lambda^2(2 - e^{-2\bar{\theta}_t})\right) \tag{20}$$

Therefore, we could rewrite the equality as:

$$\|x_Q - f_{\mathrm{OU}}(x_Q, \mu; t_0)\|$$
$$\geq \left\|\|z_T\|_2^2 - \left\|\int_T^0 -\frac{d\sigma_t^2}{dt}\nabla_x \log p_t(x)\, dt + d\overline{\mathbf{w}}_t\right\|_2^2\right\|_2^2 \tag{21}$$
$$= \left\|(x_0 - \mu)^2 e^{-2\bar{\theta}_t} + \lambda^2(2 - e^{-2\bar{\theta}_t}) - C\sigma_{\max}^4 - \left\|\int_T^0 \sqrt{\frac{d\sigma_t^2}{dt}}\, d\overline{w}_t\right\|_2^2\right\|_2^2$$

Since we require $\lambda = \frac{\sigma_t^2}{2\theta_t}$, we can find a $\sigma_{max}$ such that $\sigma_t < \sigma_{max}$ for all $t$. The last term only concerns the random noise, according to [32], we have that the last term is equivalent to the squared $L_2$ norm of a random variable from a Wiener process at time $t = 0$, with marginal distribution being $\epsilon \sim \mathcal{N}\left(\mathbf{0}, \sigma_T^2\mathbf{I}\right)$. The squared $L_2$ norm of $\epsilon$ divided by $\sigma_T^2$ is a $\chi^2$-distribution with $d$-degrees of freedom, we have the following one-sided tail bound, according to [32]

$$\Pr\left(\|\epsilon\|_2^2/\sigma^2\,(t_0) \geq d + 2\sqrt{d \cdot - \log \delta} - 2\log \delta\right) \leq \exp(\log \delta) = \delta$$

Therefore, with probability $1 - \delta$, the HQ image and retrieved image has a lower bound of,(d is the number of dimension for $x_0$),

$$| \left( (x_0 - \mu)^2 - \lambda^2 \right) \mathrm{e}^{-2\bar{\theta}_T} + 2\lambda^2$$
$$- \sigma_{max}^2 \left( C\sigma_{max}^2 + d + 2\sqrt{-d \cdot \log \delta} - 2\log \delta \right) |$$

A SDE modeling that has suffered complex degradation is considered 'corrupted':

> **Example 1.** Based on the Prop. The conditional SDE diffusion is composed of three general stages, forward, backward, and sampling. Both time-reversal processes have been declared unstable under the degradation process $\boldsymbol{\mu}$ according to the Prop. 2.

The following advantages would be offered by our measure-preserving dynamical system when formulated as SDEs:

> **Intuition 1.** A two-sided measure-preserving random dynamical(MP-RDS) system formulation enables us to use the Poincare recurrence theorem, intuitively, with a two-sided MP-RDS $\varphi_t$, the Poincare recurrence theorem ensures that the system $\varphi_t$ starts from terminal condition $x_T$, run backwards in time, will hit a region $(x_0 - \epsilon, x_0 + \epsilon)$ for small $\epsilon$ in finite time, where $x_0$ is the high-quality image.

# F    Temporal Distribution Discrepancy during Sampling

**Theorem 2** *Suppose that both the drift $b_t(x)$ and diffusion $\sigma_t(x)$ term of a stochastic process $x_t$ is Lipschitz continuous with some constant $K$, moreover, $x \in \mathbb{L}^2(\mathbb{F}, \mathbb{R})$ is a solution to the SDE*

$$dx_t = b(t, x)dt + \sigma(t, x)dW_t$$

*and initial condition $b_0, \sigma_0$ then we have that*

$$\mathbb{E}\left[|x_T - x_0|^2\right] \le C I_0^2, \tag{22}$$

*such that*

$$I_0^2 := \mathbb{E}\left[\left(\int_0^T |b_0|\, dt\right)^2 + \int_0^T |\sigma_0|^2\, dt\right]$$

*and C depends only on $T, K$, which is the running time and Lipschitz constant*

Firstly, we have the following relationship:

$$
\begin{aligned}
x_T &\le |x_0| + \int_0^T |b(t, x)|\, dt + \sup_{0 \le t \le T}\left|\int_0^t \sigma(s, x)\, dW_s\right|, \\
|x_T - x_0| &\le \int_0^T |b(t, x)|\, dt + \sup_{0 \le t \le T}\left|\int_0^t \sigma(s, x)\, dW_s\right|.
\end{aligned}
\tag{23}
$$

Squaring both sides, taking expectations, and applying the Burkholder-Davis-Gundy inequality, we get:

$$
\begin{aligned}
&\mathbb{E}\left[|x_T - x_0|^2\right] \\
&\le C\mathbb{E}\left[\left(\int_0^T |b(t, x_t)|\, dt\right)^2 + \sup_{0 \le t \le T}\left(\int_0^t \sigma(s, x_s)\, dB_s\right)^2\right] \\
&\le C\mathbb{E}\left[\left(\int_0^T [|b_0| + |x_t|]\, dt\right)^2 + \int_0^T |\sigma(t, x_t)|^2\, dt\right] \\
&\le C\mathbb{E}\left[\left(\int_0^T |b_0|\, dt\right)^2 + \int_0^T \left[|\sigma_0|^2 + |x_t|^2\right]\, dt\right].
\end{aligned}
\tag{24}
$$

**Remark:** It should be noted that the constant $C$, which depends on $T$ and $K$, varies from line to line.

Next, we show that for any $\varepsilon > 0$, there exists a constant $C_\varepsilon > 0$ such that:

$$\sup_{0 \le t \le T} \mathbb{E}\left[|x_t|^2\right] \le \varepsilon \mathbb{E}\left[|x_T^*|^2\right] + C_\varepsilon I_0^2. \tag{25}$$

Applying Itô's formula, we get:

$$d\,|x_t|^2 = \left[2x_t b(t, x_t) + |\sigma(t, x_t)|^2\right]\,dt + 2x_t \sigma(t, x_t)\,dB_t. \tag{26}$$

Considering the martingale property of the third term, integrating, and taking expectation on both sides, we have:

$$
\begin{aligned}
\mathbb{E}\left[|x_t|^2\right] &= \mathbb{E}\left[|x_0|^2 + \int_0^t \left[2x_s b(s, x_s) + |\sigma(s, x_s)|^2\right]\,ds\right] \\
&\le \mathbb{E}\left[|x_0|^2 + \int_0^t \left[C\,|x_s|^2 + 2\,|x_s|\,|b_0| + C\,|\sigma_0|^2\right]\,ds\right] \\
&\le C \int_0^t \mathbb{E}\left[|x_s|^2\right]\,ds + 2\mathbb{E}\left[x_T \int_0^T |b_0|\,ds\right] + CI_0^2.
\end{aligned} \tag{27}
$$

Using Gronwall's inequality, we can prove (25) with the result above. Substituting (25) into (24) completes the proof, showing that the distance between $x_T$ and $x_0$ in the $L^2$-sense is bounded by $C(K, T)$.

With theorem 1 in hand, since the OU process is Lipschitz continuous, then the reverse process for the OU process is also Lipschitz continuous.

Now, suppose that the Lipschitz constant for the **reverse process** is given by $\bar{K}$. then we have that in $L^2$-norm, the distance between any final state $x_T \in \mathcal{N}(\mu, \lambda)$ and the initial state(HQ image) is bounded by a constant that only depends on time $T$ and Lipschitz constant $\bar{K}$, and the initial condition for the drift and diffusion term, where we denote as $C(\bar{K}, T, \mu, \lambda)$, which will be written as $C(\bar{K}, T)$ for short.

Now, since the time $T$ is finite, where theoretically only when $T \to \infty$ would $x_T$ converge to the theoretical stationary distribution, thus, if we denote the sample from $T$ time steps by $\hat{x}_T$, and $x_T$ by the sample from the theoretical stationary distribution, then we have $\mathbb{E}[x_T - \hat{x}_T)] > \epsilon(T, K, \mu, \lambda)$, note that the $K$ here is the Lipschitz constant for the forward process, and this distance strictly decrease in $T$.

Suppose that in inference, when the ground truth $x_0$ is unknown, the distance between ground truth $x_0$ and sample from the theoretical distribution $x_T$ is bounded from below by

$$||x_0 - x_\infty|| > C(\bar{K}, T)I_0^2 + \epsilon \tag{28}$$

Then, we can see that the gap between $x_T$ and $\hat{x}_T$ increases such bound, which is

$$
\begin{aligned}
||x_0 - \hat{x}_T|| &\ge ||x_0 - x_\infty - (\hat{x}_T - x_\infty)|| \\
&> C(\bar{K}, T)I_0^2 + \epsilon(T, K, \mu, \lambda)
\end{aligned}
$$

where $T$ is the time step for the forward process, and $\epsilon$ is some strictly positive constant. Now suppose that $\hat{x}_t$ is the solution to the reverse process, which runs for $T$ periods in total, then we have that for $t \in [0, T]$, denote $x_0, \hat{x}_0$ as the original HQ image and the final state of the reverse process, respectively.

$$
\begin{aligned}
\left\|x^{Quality} - \text{OU}\left(x^{Quality}, \mu; t_0, \theta\right)\right\|_2^2 &= \\
||x_0 - \hat{x}_0|| = ||x_0 - \hat{x}_T - (\hat{x}_0 - \hat{x}_T)|| &\ge \\
|\,||x_0 - \hat{x}_T|| - ||(\hat{x}_0 - x_T)||\,| &> \\
\epsilon + C(\bar{K}, T)I_0^2 - C(\bar{K}, T)I_0^2 &= \epsilon
\end{aligned} \tag{29}
$$

Therefore, the bias created in inference depends on the LQ image $\mu$, stationary variance, Lipschitz constant, and the time steps T.

# G    Stable in Probability

**Definition 7**  *Stable in Probability*
 *Given a probability space $(\Omega, \mathcal{F}, P)$ and a standard Brownian motion $W_t$, a general form of SDE for a stochastic process $x_t, 0 \leq t \leq T$ is given by*

$$dx_t = b(t, x)dt + \sigma(t, x)dW_t \tag{30}$$

*Such that the Lipschitz condition is satisfied, for both $b(t, x), \sigma(t, x)$, A solution $x(t, \omega) \equiv 0$ is said to be stable in probability for $t \geq 0$ if for any $s \geq 0$ and $\varepsilon > 0$*

$$\lim_{x_0 \to 0} \mathbf{P} \left\{ \sup_{t > s} |x^{s,x}(t)| > \varepsilon \right\} = 0. \tag{31}$$

It says that the sample path of the process issuing from a point $x$ at time $s$ will always remain within any prescribed neighbourhood of the origin with probability tending to one as $x \to 0$. In practice, this property ensures that the perturbation from the initial state caused by a stable process is bounded for all $t$ with probability one.

For example, the OU process 4 admits a unique unconditional stationary solution provided by theorem 19, however, in this example, without specifying the value that determines the stationary variance, *i.e.*, $\sigma, \theta$. If for large $\sigma$ and a sample $\hat{x}$ from the stationary distribution of the OU process, we have that

$$\mathbb{P}(\hat{x} > \mu \pm \frac{\sigma}{2\theta}) > 0 \tag{32}$$

This means that a sample from the stationary distribution of the forward process could deviate largely from $\mu$, thus making the result no different from the traditional VE(variance exploding) diffusion models that are defined as

$$dx_t = \sigma_t dW_t$$

because the variance could be set arbitrarily large if no restriction is specified. Therefore, how should such a problem be approached most easily? The Lyapunov theorem for stability has provided an easy way, without explicitly solving the SDEs, to ensure stability just from the coefficients.

Then we provide the main theorem that ensures the stability of SDE, first, we give the definition of positive definite in the Lyapunov sense

**Definition 8**  *Let $K$ denote the family of all continuous nondecreasing functions $\mu : R_+ \to R_+$ such that $\mu(0) = 0$ and $\mu(r) > 0$ if $r > 0$. For $h > 0$, let $S_h = \{x \in R^n : |x| < h\}$. A continuous function $V(x, t)$ defined on $S_h \times [t_0, \infty)$ is said to be positive-definite (in the sense of Lyapunov) if $V(0, t) \equiv 0$ and, for some $\mu \in K$,*

$$V(x, t) \geq \mu(|x|)$$

*for all $(x, t) \in S_h \times [t_0, \infty)$*

Then we will use the convention of the Lyapunov quadratic function, such that

**Definition 9**  *Lyapunov quadratic function $V$ is given*

$$V(x_t) = x_t^T Q x_t,$$

*where $Q$ is a symmetric positive-definite matrix.*

**Theorem 3**  *The function $LV$*

$$\begin{aligned} LV(x_t) = & x_t^T Q b(t, x_t) + b(t, x_t)^T Q x_t + \\ & \sigma(t, x_t)^T Q \sigma(t, x_t), \end{aligned} \tag{33}$$

*is negative-definite in some neighbourhood of $x_t = 0$ for $t \geq t_0$, with respect to system 7. Then the trivial solution of equation 7 is stochastically asymptotically stable.*

Since this theorem is important and the proof will be intuitive in explaining why such a condition could ensure stability, the proof will be put here.

**Proof:**

First, we compute $dV(x)$, which is the instantaneous growth of the Lyapunov quadratic function, gives

$$
\begin{aligned}
dV\left(x_t\right) = &V\left(x_t + dx_t\right) - V\left(x_t\right) \\
= &\left(x_t^T + dx_t^T\right) Q\left(x_t + dx_t\right) - x_t^T Q x_t \\
= &x_t^T Q b\left(t, x_t\right) dt + x_t^T Q \sigma\left(t, x_t\right) dB_t + \\
&b\left(t, x_t\right)^T dt Q x_t + \sigma\left(t, x_t\right)^T dB_t Q x_t + \\
&\sigma\left(t, x_t\right)^T Q \sigma\left(t, x_t\right) dt
\end{aligned}
$$

Then, take expectation, we can get

$$
\begin{aligned}
E\left\{dV\left(x_t\right)\right\} = &x_t^T Q b\left(t, x_t\right) dt + b\left(t, x_t\right)^T Q x_t dt + \\
&\sigma\left(t, x_t\right)^T Q \sigma\left(t, x_t\right) dt \\
= &LV\left(x_t\right) dt
\end{aligned}
$$

Then, if we assume that $LV(x)$

$$
-LV\left(x_t\right) \geq kV\left(x_t\right)
$$

such that $k$ is a constant, then

$$
\begin{aligned}
&\frac{d}{dt} E\left\{V\left(x_t\right)\right\} \leq -k E\left\{V\left(x_t\right)\right\}, \\
&E\left\{V\left(x_t\right)\right\} \leq \exp(-kt).
\end{aligned}
\tag{34}
$$

As can be seen from the proof, the operator $LV$ as the function of the SDE $x_t$ is the expectation of the $dV(x_t)$, and the negative semi-definiteness can be regarded as requiring $dV(x_t)$ to be a contraction. This can be understood from 34 such that

$$
\frac{d}{dt} E(V(x_t))/E(V(x_t)) < -k
$$

for $k > 0$

# H    Implementation details

**Model Implementation:** Our exploration into mitigating Temporal Distribution Discrepancy in diffusion models employs two neural network architectures, each catering to different dataset complexities.

We utilize the adopted UNet, a staple in DDPM [24] and DDIM[53] frameworks, chosen for its widespread use and strong benchmarking capabilities. By solving the discrepancy through this established structure, we achieve state-of-the-art results on synthetic data, showcasing the potential of improving transanary SDE diffusion models in terms of Temporal Distribution Discrepancy and stationary process. Additionally, the use of this prevalent architecture allows for comprehensive analysis and discussion.

Real-world data with its inherent complexity, such as combined degradations, large resolutions, and extensive interdependencies, requires an architecture beyond the conventional UNet. Our improved model incorporates Squeeze-and-Excitation [38] and NAF [8], explicitly designed to capture intricate feature interrelations. While these models do not seek to innovate the architectural paradigm, it provides a solid baseline that provides better feature extraction ability than UNet performance in demanding scenarios.

**Additional details:** The U-Net we adopted is similar to DDPM as described in [37, 11, 9], where the improved model incorporates Squeeze-and-Excitation [38] to replace the Attention module within NAF [8]. EDSR [33] is employed as the base model for TTA-based comparison methods in MRI super-resolution. In different downstream tasks, we follow the common setting of the latest compared methods: Deraining [37], Real Dehazing [47], MRI Reconstruction [26], and MRI super-resolution [14] and [33]. Below are more specific details about MRI reconstruction and MRI

Table 9: Datasets parameters and split setting. IOP details of the scan parameters are not available.

| Datasets | *Data setting* | | *Collection* | | | *Sequence parameters* | | |
| *Source domain* | Subjects | Slices | Hospitals | Scanner | Repetition time | Echo train length | Matrix size | Receiver coil |
| HH (IXI Brain) | 184 | 60 | Hammersmith | Philips 3T | 5725 | 16 | 192 x 187 | Single |
| *Target domain* | | | | | | | | |
| Guys (IXI Brain) | 30 | 60 | Guy's Hospital | Philips 1.5T | 8178 | 16 | Unnormalized | Single |
| IOP (IXI Brain) | 30 | 60 | Institute of Psychiatry | GE 1.5T | | Unknown | | Single |

| Method | PSNR↑ | SSIM↑ | LPIPS↓ |
| --- | --- | --- | --- |
| WD UNet | 27.31 | 0.8322 | 0.227 |
| SN UNet | 28.63 | 0.8687 | 0.144 |
| UNet | 31.03 | 0.9001 | 0.058 |

Table 10: Evaluation of the Lipschitz continuity.

superresolution. For most of the experiments, the training patch-size is set to 128x128 with a batch size of 16. We utilize the Adam optimizer with $\beta_1 = 0.9, \beta_2 = 0.999$, a learning rate of $10^{-4}$ with a decay strategy. Our models are trained on three RTX 6000 GPUs for about four days, each with 40GB of memory. Random seed is 42. All mathematical variants follow the cosine schedule as per [42]. The variance $\lambda$ is set to 10 for the OU and stationary processes. In the coef. decoupled SDE, $\sigma$ is kept decoupled and does not vary with $\theta$. We observed that the adaptability of $\tau$ to tasks is contingent upon the task's corruption for model stability and generalizability.

$\tau$ is a hyperparameter as one of the ways to introduce Measure-preserving Dynamics to shape more stable SDE diffusion. It is informed by the deviation between the degraded image and the expected high-quality image. For tasks with moderate intra-domain deviations, $\tau$ is set to 2, while for tasks with substantial cross-domain and degradation discrepancies, a larger $\tau$ is used. This metric is based on RDS settings of different degradations, rather than learned. A MLP can be used to learn $\tau$ as adaptive metric, although it is not comparable with obvious improvements.

**SGM, and Transitionary SGMs *vs.* D$^3$GM:** We perform qualitative 2 and quantitative 11 analyses using variants of closely related formulations for Prop. 1 and 2 and evaluate across (A) SGMs and (B) transitionary SGMs. (A) uses a common score-based SDE, (B) uses a Coefficient Decoupled SDE (*e.g.*, variance exploding SDE with the drift term $\mu$) according to Prop. 1 and OU SDE, alongside our D$^3$GM.

**MRI Reconstruction:** We applied 584 proton-density weighted knee MRI scans without fat suppression. These were subsequently partitioned into a training set (420 scans), a validation set (64 scans), and a testing set (100 scans). For each scan, we extracted 20 coronal 2D single-channel complex-valued slices, predominantly from the central region with the uniform size of 320 x 320. Our test set differs from the 200 reported in [26], possibly as a result of the change in the official versions of FastMRI. We subjected all experiments to Cartesian under-sampling masks with undersampling factor 8x and 16x. In the undersampled MRI scans, the acquisition process entails sampling a fractional subset of the Fourier-space (k-space), typically governed by a mask along the dimension of undersampling rate. Undersampling inherently induces aliasing artifacts in the resultant images. Considering the domain deviation, we did not supplement the extra test set into the final 100 samples. Otherwise, we stayed consistent with the details of the paper. For a robust comparison, we benchmarked against a diverse set of deep learning-based state-of-the-art reconstruction methods, including CNN-based approaches such as D5C5 [51] and DAGAN [62], the Transformer-based SwinMR [27], diffusion model-inspired DiffuseRecon [44], and the CDiffMR [26]. Quantitative

Table 11: Exemplary deraining results of (A) SGM, (B) transitionary SGMs: Coefficent Decoupled SDE, OU SDE, and D$^3$GM.

| | Method | PSNR↑ | SSIM↑ | LPIPS↓ |
| --- | --- | --- | --- | --- |
| A | SGM | 27.27 | 0.840 | 0.144 |
| | Coef. Dec. SDE | 26.18 | 0.826 | 0.205 |
| B | OU SDE | 30.58 | 0.900 | 0.051 |
| | D$^3$GM (ours) | **32.41** | **0.912** | **0.040** |

results in Tab. 5 exhibit a differing trend from the image domain. The task-specific diffusion models achieve better results and are more capable of capturing the complex degradation that occurs in the frequency domain.

Generally, MRI uses a mask in the phase-encoding direction (the shortest anatomical direction [41]) to model the complex degradation caused by undersampling. In the knee data, based upon the uncertainty of clinical diagnosis (longitudinal artifacts can significantly confuse the diagnosis of meniscal injury), we also masked the frequency direction, which will cause resolution reduction, deterioration of image features, and longitudinal artifacts, more qualitative results can be found in the next section.

**MRI Super-resolution:** IXI[4] contains clinical T1- and T2-weighted scans from three hospitals with different imaging protocols: HH, Guys, and IOP. We selected 184 HH T2 subjects as the source-domain data [train/val/test ratio: 7:1:2], and 30 subjects each from Guys and IOP as two target-domain datasets without degradation, acquisition parameters, datasets, and patient-wise crossovers. The central 60 slices were selected.

We consider two benchmark ideas: (1) Blind SR degradation methods in frequency SFM [19] and spatial PDM [70] domain. (2) Source-free SR adaptation ACT [14] using external priors and test-time adaptation proposed initially for MRI CST [14] with data consistency in the source-domain. We are concerned that existing work on robustness and generalization focuses on medical image segmentation and natural images, and rarely on super-resolution and reconstruction of medical images. Employing the available evaluation criteria in this inevitable problem in practice makes it difficult to reflect the performance of D$^3$GM. Therefore, domain-aware datasets isolation standard for public datasets is designed for the SRR adaptation to cross-domain data rather than same-domain in a source-free manner. The publicly available data were split into several subsets based on hospitals, scanner, acquisition parameters, modality, and anatomy as illustrated in Table 9. Explicit reference standards implicitly correspond to various degradation patterns, thus enabling the isolation of natural degradation patterns in source training domain and target-testing domain. In addition to this, different artificial degradation patterns for the subsets in the training and testing domains are employed: K-space truncation downsampling was applied to obtain LR data in the source domain and a kernel degradation [5] was applied in target domain.

We reproduce two types of benchmark ideas on the top of EDSR [33] backbone to achieve the multi-purpose goal: (1) *Repurposed blind SR (BSR) for cross-domain data*: We utilized BSR degradation methods in frequency **SFM** [19] and spatial **PDM** [70] domain. (2) *Test-time adaptation (TTA)*: We compare to a source-free TTA method **ACT** [14] using external priors and a second TTA method proposed initially for MR reconstruction **CST** [14] with cycle consistency at source-domain. The settings and adaptation strategies of the comparison methods were used directly.

**Lipschitz Continuity for Stationary Process:**

We hypothesize that ensuring Lipschitz continuity of the neural network is pivotal for the convergence and stability of the diffusion process, particularly in the context of achieving a stationary process. Theoretically, Lipschitz continuity offers two key benefits: (1) it mitigates the impact of small perturbations in input data or model parameters, thus safeguarding against excessive output variability which could cause numerical instabilities or result in an ill-posed problem, and (2) it guarantees the existence of a unique solution to the diffusion process, underpinning the reliability and convergence of the numerical methods employed to solve these equations.

To instantiate these theoretical benefits within our architecture, we integrated spectral normalization (SN) [40] and weight decay (WD) [35] into a U-Net structured score network. In the case of SN, it is achieved by rescaling each layer's weight matrix $W^{(l)}$ by its spectral norm, $\sigma_{max}(W^{(l)})$, to obtain a normalized weight $\tilde{W}^{(l)} = \frac{W^{(l)}}{\sigma_{max}(W^{(l)})}$. By doing so, we intend to control the overall Lipschitz constant of the network for robust score matching within our diffusion model framework. A quantitive result can be seen in Tab. 10.

---

[4]http://brain-development.org/ixi-dataset/

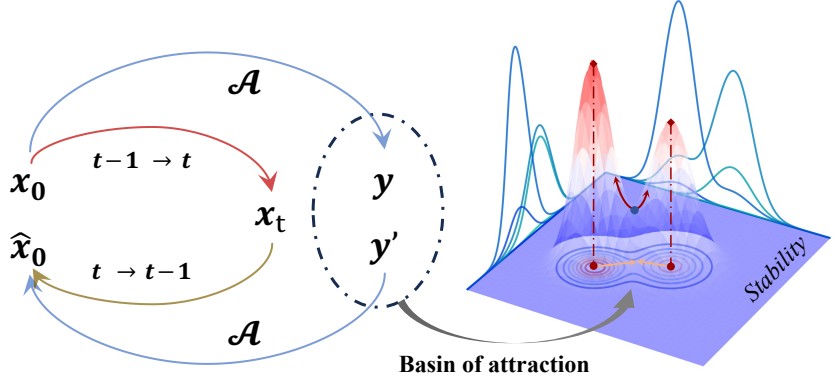

Figure 4: Reverse Initialization with Basin of attraction.

# I Additional Insights

**Basin of Attraction in Reverse Sampling:** Within our diffusion framework, we establish a forward process with variety to accommodate a wide range of potential corruptions, ensuring the desired final distribution close to the expected distribution. We consider that our diffusion models inherently encompass the forward operator $\mathcal{A}$ within their structure. The transition from high to low-quality images is implicitly encoded in the diffusion pathway, hence the additional forward operator guidance might not contribute supplementary information, which we initially hypothesized would enhance the reverse process.

Thus, the keypoint transfers from the $\mathcal{A}$ to the initial $y'$ (Detailed analysis based on Prop. 2 is provided in Appx. F). Also we found this problem fall into the Basin of Attraction (BA) in dynamical system as shown in Fig. 4. BA can be interpreted as the quality of the attractor of the degraded image in the reverse process here. A BA-guided initialization might be a more effective approach for posterior sampling in transitional SDE diffusion models. By initializing the reverse process closer to the expected solution, we may bypass the initial hurdle of distribution mismatch.

# J More Qualitative Results

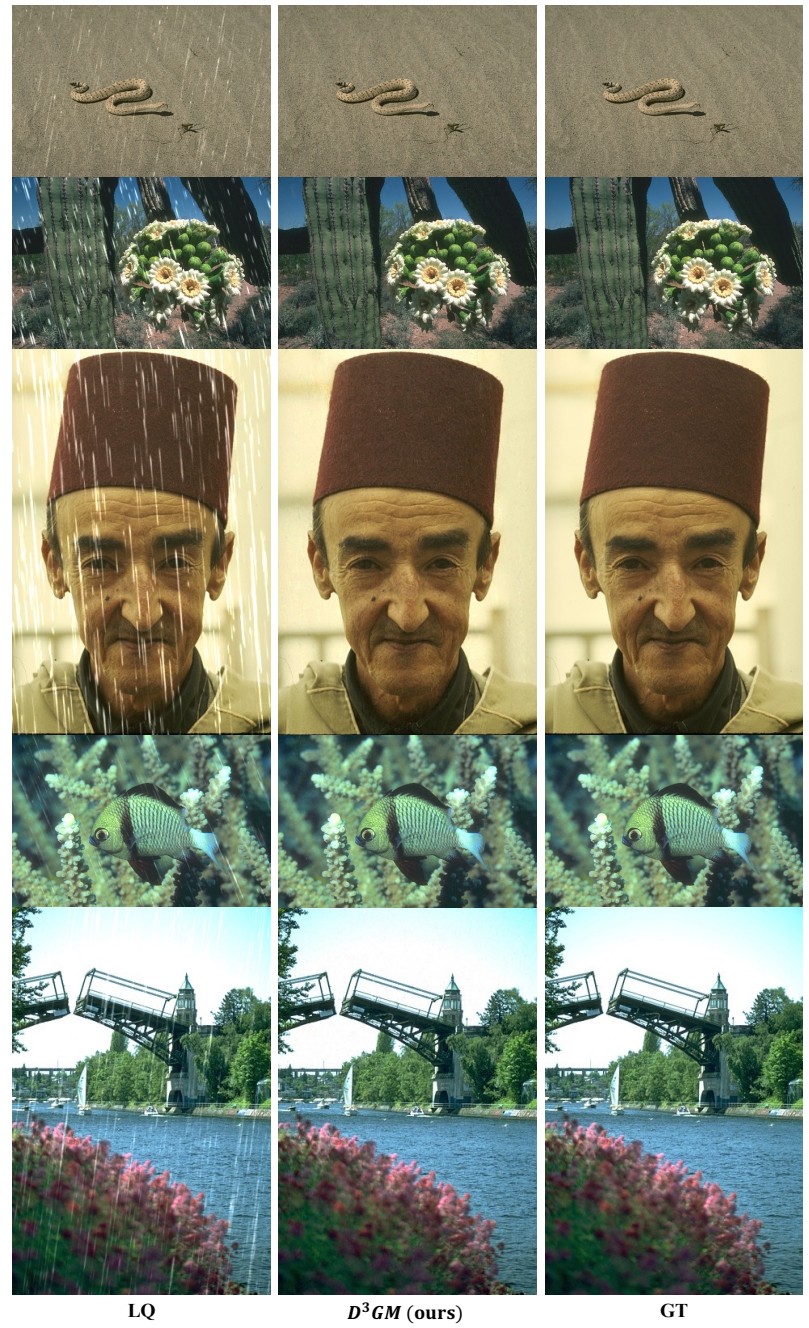

| LQ | $D^3GM$ (ours) | GT |

Figure 5: Deraining results with light rain images of our method.

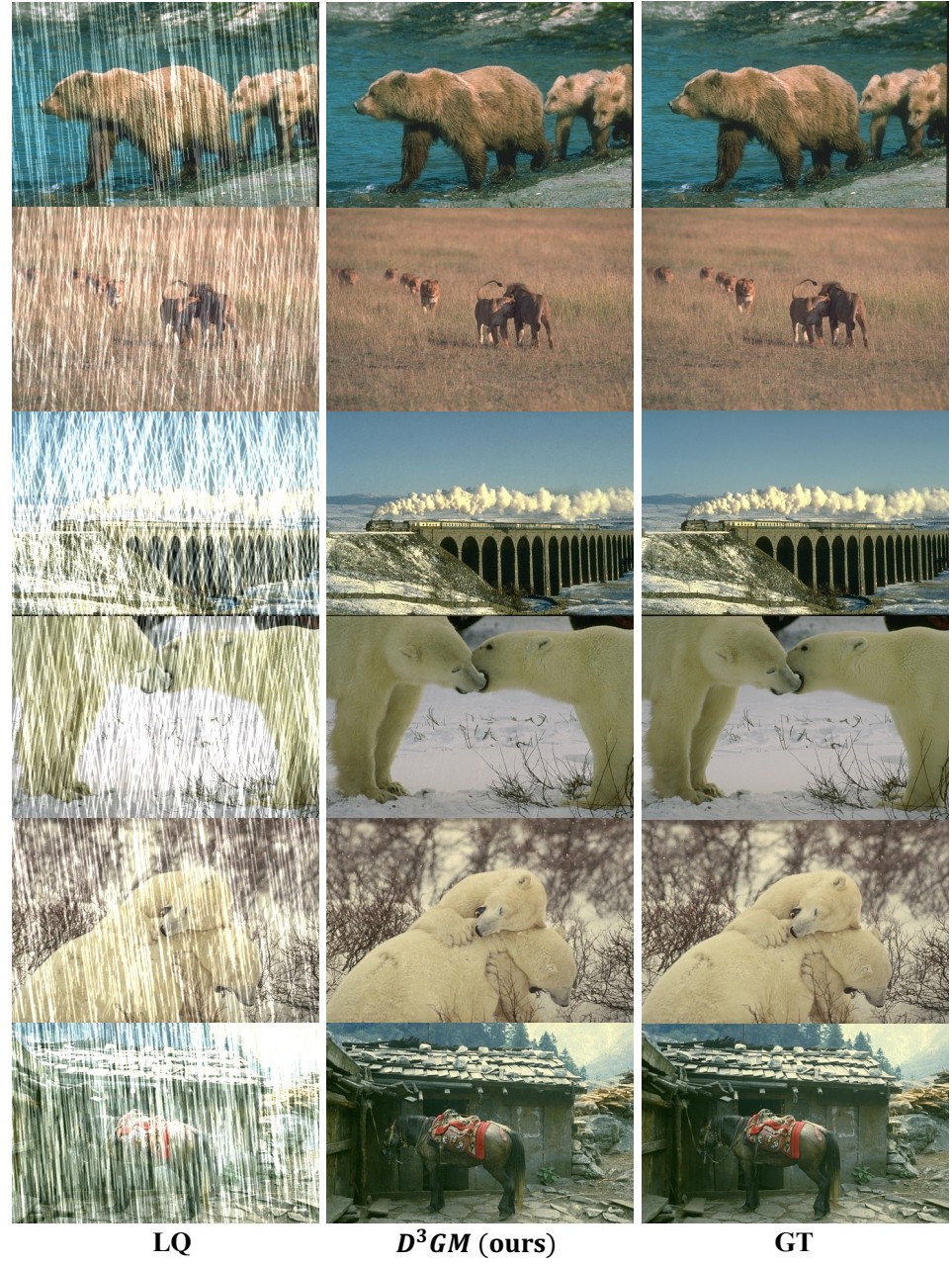

**LQ**       $D^3GM$ **(ours)**      **GT**

Figure 6: Deraining results with heavy rain images of our method.

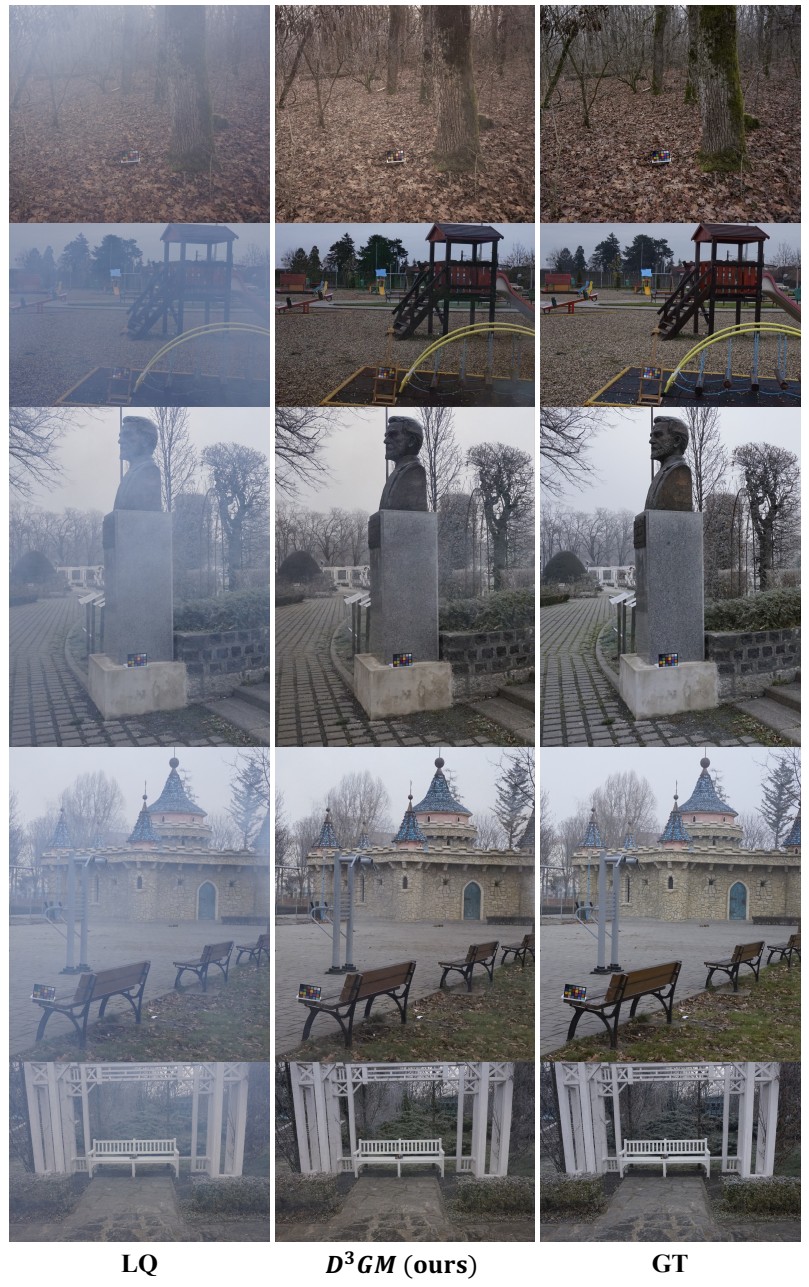

| LQ | $D^3GM$ (ours) | GT |

Figure 7: Dehazing results with real hazy images of our method.

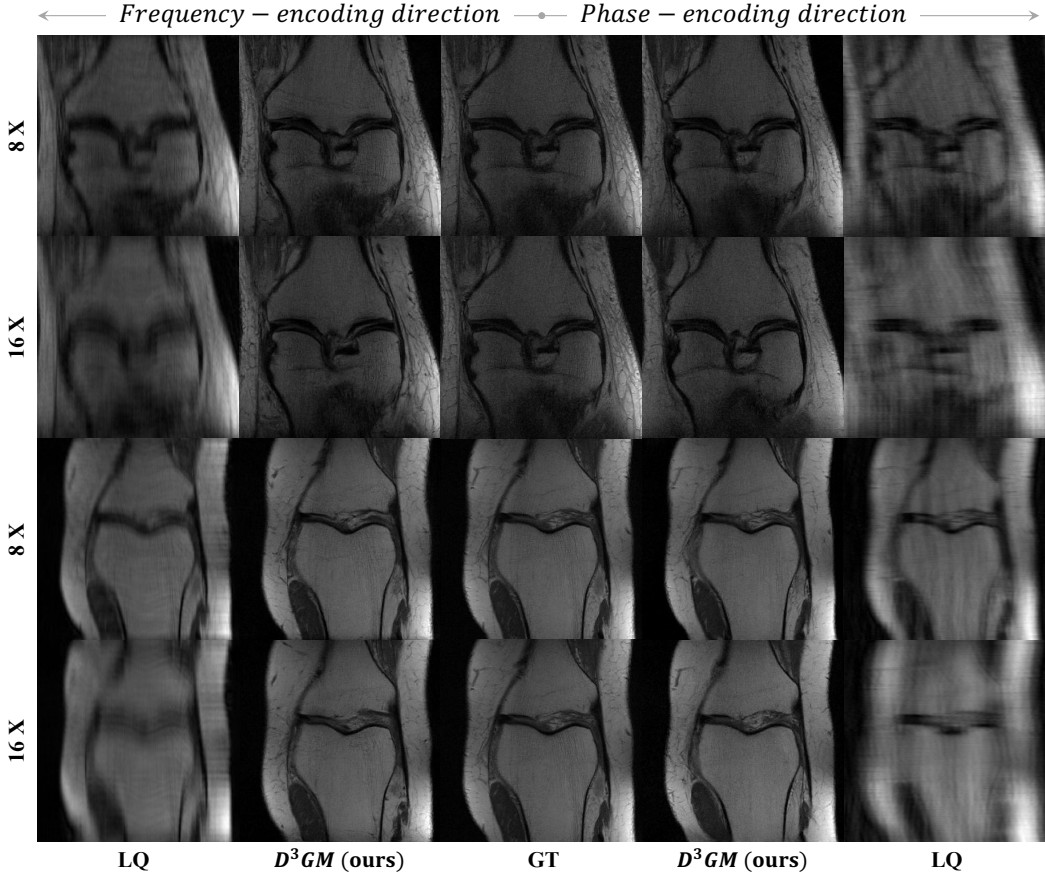

Figure 8: MRI reconstruction results with undersampling rate x8 and x16, on Frequency-encoding and Phase-encoding directions.

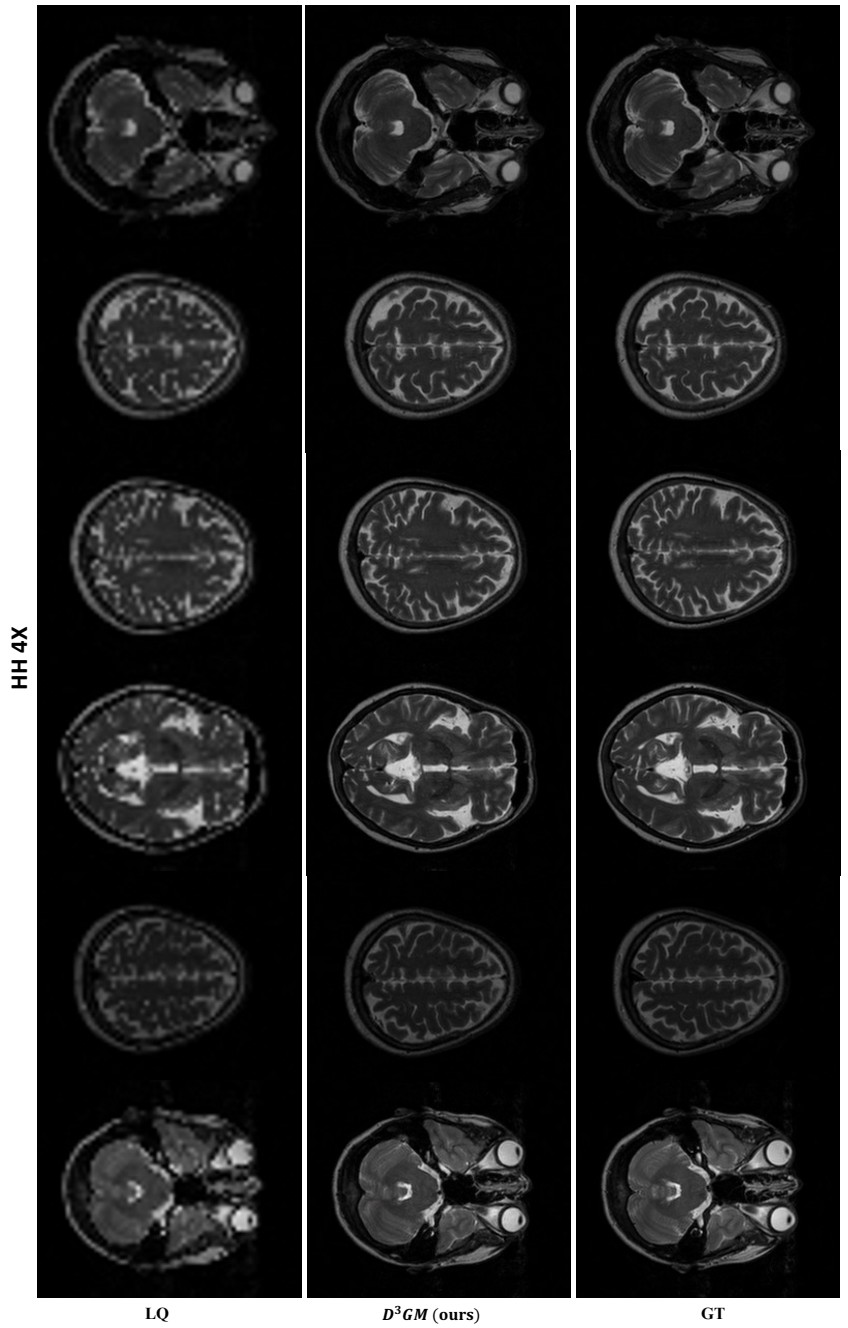

Figure 9: MRI super-resolution results with in-domain images of our method.

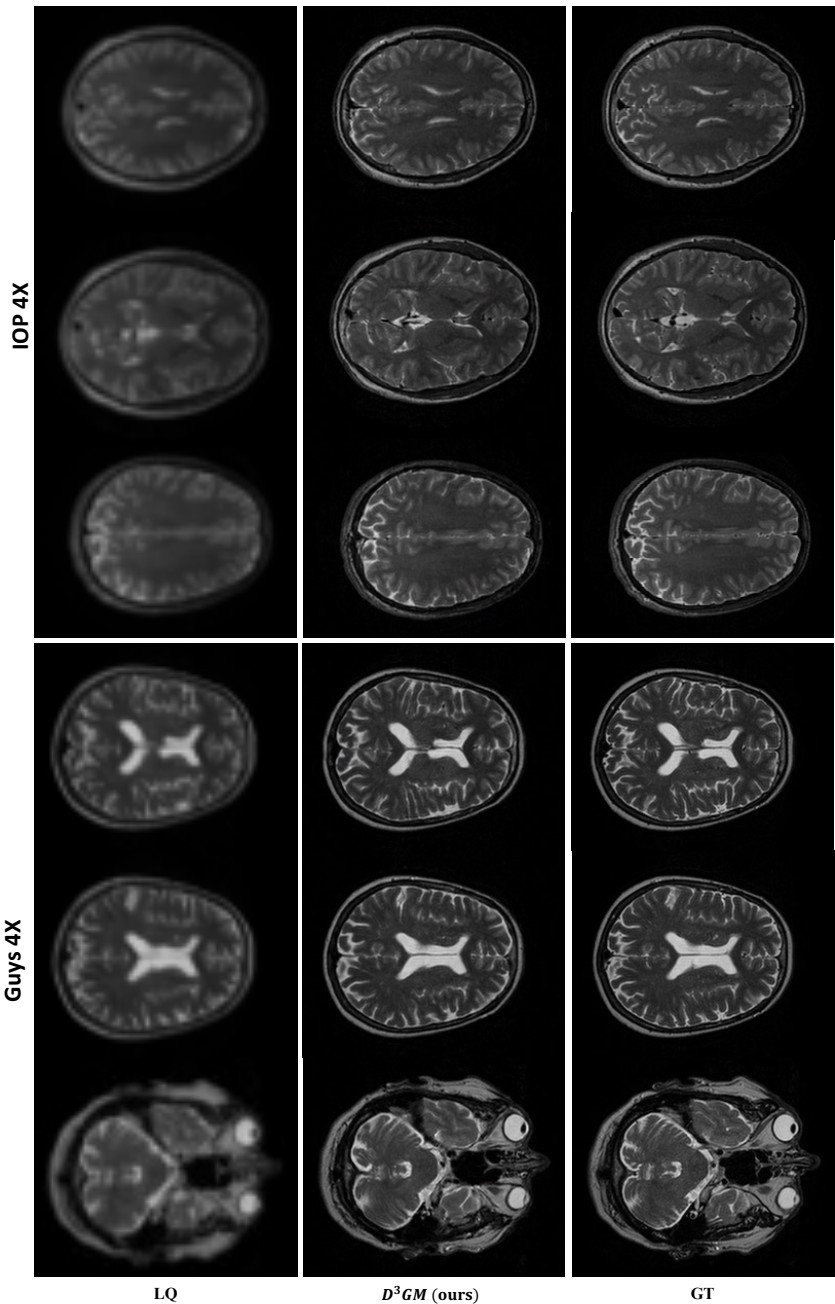

Figure 10: MRI super-resolution results with cross-domain (different imaging devices and degradation methods) images of our method.

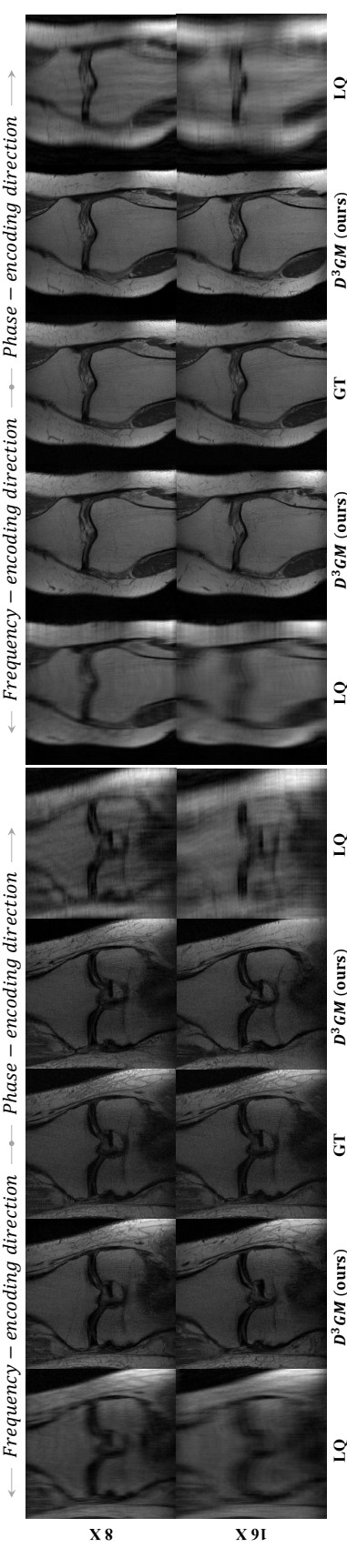

Figure 11: MRI reconstruction results with undersampling rate x8 and x16 of our method, on Phase-encoding and Frequency-encoding directions.

