# OpenReview forum: "Stability and Generalizability in SDE Diffusion Models with Measure-Preserving Dynamics"
_NeurIPS.cc/2024/Conference — NeurIPS 2024 poster_

### Official Review · Reviewer_z6fz · 2024-06-26

**Soundness:** 4
**Presentation:** 3
**Contribution:** 3
**Rating:** 7
**Confidence:** 4

**Summary:**

The author(s) of the paper provides a theoretically sound method of a Dynamics-aware SDE Diffusion Generative Model (D^3GM) to enhance the stability and generalizability of inverse problem diffusion models. The authors provide a rigorous mathematical examination of the temporal distribution discrepancy for the instability issue of transitionary score-based generative models. The analysis extends the traditional Ornstein-Uhlenbeck (OU) process to random dynamical systems (RDS), focusing on the stability of SDE. The authors then proposed a novel method (D^3GM) that combines the stationary process to relieve the temporal distribution discrepancy problem following the measure-preserving dynamics from RDS; this method could guide the SDE diffusion to a desired stable solution. The experimental results from the authors also indicate the efficiency of D^3GM under different situations.

**Strengths:**

1. The authors provide a novel method (D^3GM) integrating measure-preserving dynamics into SDE diffusion models. This is the fundamental contribution of this paper.
2. Extending the Ornstein-Uhlenbeck process to a random dynamical system is innovative, allowing the community to better understand the instability problems of diffusion models. This paper also provides a detailed mathematical measurement of the discrepancy between the reference and the retrieved data.
3. The logic of this paper is clear.

**Weaknesses:**

1. The paper is theoretically sound, but it might be difficult for readers with no math background to follow. Consider adding more explanations for the theoretical part.
2. See below

**Questions:**

1. Although this paper has solid theoretical support, the experiment is not as good as the theoretical part. It would be helpful if the authors could add more visual comparisons between these models (I noticed Figure 3, but from the set of results, there seems to be no significant improvement). This would help readers get a clear understanding of the proposed method's exact performance compared with other methods.
2. at line 217, the COS is chosen to balance 'the trade-off between complexity and effectiveness'. This is unclear; it will be helpful to add more theoretical support or a set of experiments to empirically show COS is better than other options.

**Limitations:**

Even though the paper fulfills the theoretical gap, the actual improvement of the model performance seems limited.

---

> ### Author Rebuttal · Authors · 2024-08-05
>
> We thank Reviewer z6fz for the excellent summary of our paper and for capturing our core contributions so well. We appreciate the positive comments on our clear logic and detailed mathematical analysis.
>
> >Q. Add explanations for non-math background readers:
>
> We acknowledge that combining theoretical knowledge across multiple domains is challenging. To address this, we have included all relevant mathematical theories and derivations in the appendix. Additionally, we have provided Intuition 1 (Line 142) and Intuition 2 (L. 158), as well as Example 1 (L. 143) and Example 2 (L. 200), at key points to aid understanding. Furthermore, we used a more vivid analogy in Appendix B (L. 480, P. 14) to guide readers through the connection between measure-preserving dynamics and inverse problems.
>
> Besides these enhancements, we will include additional explanations and illustrative examples in the camera-ready version to make the concepts more accessible. Specifically, we will also add more intuitive descriptions and diagrams to bridge the gap between the rigorous mathematical framework and practical understanding.
>
> >Q. Need for more visual comparisons:
>
> We thank the reviewer for highlighting the importance of Figure 3 and other visual comparisons. In the revised version, we will expand Figure 3 to include more comparative visuals against other baseline methods.
>
> We will focus more on visual comparisons with state-of-the-art (SOTA) methods. Given time and space constraints, we have included several visual comparisons with SOTA methods in the provided rebuttal PDF.
>
> >Q. Clarification on COS:
>
> We appreciate the reviewer's comment regarding the use of Cos at Line 217. We have provided visual comparisons in the rebuttal PDF. The visual results and discussions in Q5 of the general rebuttal will be incorporated into the final version and appendix.
>
> >Q. Limitation on the model performance:
>
> We understand the reviewer's observation regarding the limited apparent improvement in model performance. Our performance primarily lies in the robustness and generalization of resulting models, especially when applied to real-world data. Given the inherently challenging nature of these tasks, achieving practical usability is in itself a significant contribution.
>
> Our method not only demonstrates exceptional stability but also proves its extensive applicability across various practical scenarios, such as MRI reconstruction, dehazing, and deraining. Compared to other task-specific state-of-the-art methods, our approach excels with dramatically reduced FLOPs and complexities in Table 8 (L. 321, P. 9).
>
> We have provided comparisons with other advanced methods in Table 7 (L. 321, P. 9), and additionally included examples in the rebuttal PDF highlighting where these advanced methods fail to handle real-world degradations.
>
> It is a pleasure to have our innovative approach and clear logic recognized by you. The theoretical parts will be made more accessible by adding more visual comparisons and explanations. The suggestions you provided have greatly enhanced the completeness of our work.

---

> > ### Comment · Reviewer_z6fz · 2024-08-12
> >
> > Thank you so much for your rebuttals. I think most of them make sense to me.

---

### Official Review · Reviewer_J29t · 2024-07-12

**Soundness:** 3
**Presentation:** 2
**Contribution:** 2
**Rating:** 5
**Confidence:** 3

**Summary:**

The paper addresses the use of diffusion models in solving inverse problems, which involve estimating causal factors from degraded data. Traditional methods often fall short in real-world scenarios due to accumulated errors and biases. To tackle these issues, the authors propose a new theoretical framework based on measure-preserving dynamics of Random Dynamical Systems (RDS) for Stochastic Differential Equation (SDE) diffusion models. They introduce the Dynamics-aware SDE Diffusion Generative Model (D3GM), which enhances the stability and generalizability of these models. Experimental results, particularly in magnetic resonance imaging (MRI), demonstrate the framework’s effectiveness.

**Strengths:**

Solving inverse problems is critical in many real-world applications. Discussing the problem from the measure-preserving dynamical system perspective is interesting.

**Weaknesses:**

1. It is unclear to me how this work is different from DDBM [1] and Augmented bridge matching [2]. In fact, DDBM's setting is more general by considering diffusion bridges derived from h-transform. That setting covers OU processes (with some reparameterization on t if necessary).

2. The empirical study does not include an evaluation on the sampled image quality. (It is mentioned at L245 that FID will be reported; however, I did not find any in the paper. )

3. The connection between the theoretical work in Sec 3 and the implementation in Sec 4 is unclear.

4. There are multiple choices of drift and diffusion coefficients. However, there are no discussions/ablation studies to show how to choose them.

5. There is no performance comparison with similar implementations like DDBM and I2SB.

[1] Denoising Diffusion Bridge Models, Linqi Zhou, Aaron Lou, Samar Khanna, Stefano Ermon, 2023
[2] Augmented bridge matching, Valentin De Bortoli, Guan-Horng Liu, Tianrong Chen, Evangelos A Theodorou, Weilie Nie, 2023

**Questions:**

I have mentioned several problems in the Weakness section. In addition,

1. For the training of NN, e.g. dehazing, you mentioned there were only 100 pairs images for training and testing. Is the model barely trained with this much data?

**Limitations:**

I am not aware of any potential negative societal impact of this work.

---

> ### Author Rebuttal · Authors · 2024-08-05
>
> We would like to thank Reviewer J29t for your valuable feedback and insightful comments. Your suggestions have significantly contributed to highlighting the distinctiveness and clarity of our work.
>
> >Q. Difference from DDBM and Augmented Bridge Matching:
>
> We thank the reviewer for mentioning these two works. We recognize that DDBM [1] and Augmented Bridge Matching(ABM) [2] (close-source) were not published at the time of our submission. We will include them in the introduction and discussion sections of the final paper. Despite this, our approach remains distinctly different:
>
> Both DDBM and ABM rely on the theoretical foundation of diffusion bridges and Doob’s h-transform. The h-transform is an almost sure path between two endpoints where DDBM fixes it to be the forward process and learns the reverse. The learned generalized time-reversal ODE is designed for paired image translation, as stated at the end of DDBM’s section 4 [1]. For image restoration tasks with complex degradations unseen during training (i.e., $(\tilde{x}\_{hq}, \tilde{x}\_{lq}) \notin q\_{data}(x,y)$), the DDBM ODE cannot generalize to capture the correct reversal, as shown in rebuttal PDF. DDBM can be challenging to scale due to the reliance on a fixed forward process during training as noted by the authors [7].
>
> ABM faces a similar issue. Despite augmenting the drift with initial sample information to preserve coupling information, it loses the Markovian property and imposes more restrictions in the time-reversal process, making it unsuitable for image restoration tasks with unseen degradations.
>
> In contrast, D3GM leverages measure-preserving dynamics of Random Dynamical Systems (RDS) to enhance stability and generalizability. Unlike DDBM, which is constrained by its reliance on h-transforms, D3GM’s measure-preserving property allows distributions to revert to their original state despite complex degradations, ensuring robustness and accuracy in non-linear and non-Gaussian scenarios.
>
> >Q. Connection between theoretical work and implementation:
>
> The measure-preserving property ensures that despite complex degradations, the distribution can revert to its original state, maintaining stability and generalizability. This is achieved through the lens of Random Dynamical Systems (RDS), which provide a robust framework for analyzing temporal distribution dynamics and ensuring consistency in the diffusion process.
> We have provided more explanations in the Q1 and Q2 of the general rebuttal above and will add more detailed explanations and illustrative examples to clarify this connection in the revised version.
> >Q. Choices of drift and diffusion coefficients:
>
> We appreciate your comments on the coefficients and schedules. We have provided visual comparisons in the rebuttal PDF. The visual results and discussions in Q5 of the general rebuttal will be incorporated into the final version and appendix.
>
> >Q. Training with limited data:
>
> Obtaining paired datasets for real-world corruption scenarios is challenging. To address the limitation of having fewer pairs of images, we employ augmentations on the O-haze (2833×4657 pixels) and Dense Haze (1200×1600 pixels) datasets, adhering to common settings used in previous dehazing works [4, 5, 6]. We enhance the training dataset by randomly cropping patches from these images for each epoch, ensuring that the patches differ each time. This effectively creates a much larger dataset from the small available set, improving the robustness and generalization of our model, as demonstrated in other studies. We will clarify this in the final version and add more details in the appendix.
>
> >Q. Comparison with DDBM and I2SB
>
> Regarding I2SB, we have already compared our approach to I2SB in Table 1 (theoretically, L. 90, P. 3), Figure 2 (quantitatively, L. 248, P. 7), and Table 7 (quantitatively, L. 321, P. 9), discussing various perspectives in our submission. Additional experimental results have been included in the rebuttal PDF to further illustrate our model's performance.
> We recognize that DDBM was not published at the time of our submission. DDBM shares many similarities with I2SB, and we have provided a specific analysis in the general rebuttal's Q1 and your Q1. Thank you for mentioning these.
>
> >Q. Negative societal impact of this work:
>
> We would like to clarify that our paper does mention the potential negative societal impacts in appendix A(L. 473, P. 14). We will ensure this point is further clarified in the final version to avoid any misunderstandings.
>
> >Q. Evaluation of FID:
>
> Our paper mistakenly mentioned reporting FID, which was an oversight. We appreciate you pointing this out and will correct it in the final version. Given our contributions, adding FID would not offer additional insights, as PSNR, SSIM, and LPIPS evaluate accuracy and perceptual quality comprehensively. Additionally, we have provided detailed comparisons with state-of-the-art methods, covering practical aspects (datasets, FLOPs, complexity) and theoretical aspects (foundations, mathematical formulations).
>
> Thank you once again for your insightful feedback. Your comments have been crucial in refining our work and ensuring its robustness and clarity. We will incorporate the results, discussion and provide further explanations in the final version and appendix to ensure a comprehensive understanding.
>
> [1 ]Denoising Diffusion Bridge Models, L. Zhou, et al., ICLR, 2024
>
> [2] Augmented Bridge Matching, V. De Bortoli, et al., arXiv, 2023
>
> [3] I2SB: Image-to-Image Schrödinger Bridge, G.-H. Liu, et al., ICML, 2023
>
> [4] Mb-Taylorformer: Multi-branch Efficient Transformer Expanded by Taylor Formula for Image Dehazing, Y. Qiu, et al., ICCV, 2023
>
> [5] Image Dehazing Transformer with Transmission-Aware 3D Position Embedding, C.L. Guo, et al., CVPR, 2022
>
> [6] Single Image Dehazing for a Variety of Haze Scenarios Using Back Projected Pyramid Network, A. Singh, et al., ECCV Workshops
>
> [7] https://github.com/alexzhou907/DDBM/issues/3

---

> > ### Comment · Reviewer_J29t · 2024-08-12
> >
> > I thank the authors for the detailed replies. While the authors have argued that the proposed methods differ greatly from DDBM/I2SB, I still think many overlaps exist between them. In addition, though DDBM was published in 2024, it was uploaded to arxiv in Sep 2023.
> >
> > Regardlessly, considering that the submission made some solid theoretical contributions, I am raising the score to 5.

---

### Official Review · Reviewer_APgH · 2024-07-28

**Soundness:** 3
**Presentation:** 3
**Contribution:** 3
**Rating:** 7
**Confidence:** 3

**Summary:**

Given that existing diffusion models are limited to linear inverse problems, this paper proposes to use measure-preserving dynamics of random dynamical systems to formulate a theoretical framework for SDE diffusion models. They uncover several strategies that inherently enhance the stability and generalizability of diffusion models for inverse problems and introduce a score-based diffusion framework, D3GM. The measure-preserving property can return the degraded measurement to the original state despite complex degradation with the RDS concept of stability. Experiments on multiple restoration and reconstruction tasks, such as dehazing, deraining, and MRI reconstruction, demonstrate the stability and generalizability of the proposed D3GM framework.

**Strengths:**

- The theoretical results are interesting and open up many potential paths for future investigations.

- Clearly explains the advantages of measure-preserving dynamics in SDE diffusion, and how they motivate algorithmic design.

- For some challenging applications, such as MRI super-resolution, the derived model shows excellent generative capabilities and outperforms some well-known baseline methods.

- The writing is clear and the paper is well structured.

**Weaknesses:**

- The connection between the measure-preserving property and the proposed D3GM is not well-explained, and how the temporal distribution discrepancy is mitigated within D3GM is not intuitive.

- Why choose the perspective of measure-preserving dynamics of random dynamical systems? What unique advantages does it offer in solving challenging inverse problems? Besides the related instability analysis, are there more intuitive explanations available?

- The baselines used for comparison in the dehazing and deraining tasks are somewhat outdated, which raises questions about the real-world effectiveness of the proposed methods when compared to state-of-the-art approaches.

- Strictly speaking, although the paper provides some theoretical insights, it does not systematically resolve the issues, and the intuitions gained appear somewhat heuristic.

**Questions:**

Please refer to the weaknesses.

**Limitations:**

Yes

---

> ### Author Rebuttal · Authors · 2024-08-05
>
> We thank Reviewer APgH for the detailed review and recognition of our theoretical contributions and innovative use of measure-preserving dynamics.
>
> >Q. Connection between measure-preserving property and D3GM:
>
> The measure-preserving property ensures that despite complex degradations, the distribution can revert to its original state, maintaining stability and generalizability. This is achieved through the lens of Random Dynamical Systems (RDS), which provide a robust framework for analyzing temporal distribution dynamics and ensuring consistency in the diffusion process.
>
> We have provided more explanations in Q1, Q2 and Q3 of the general rebuttal and will add more detailed explanations and illustrative examples to clarify this connection in the revised version.
>
> >Q. Choice of measure-preserving dynamics:
>
> Unlike traditional linear approaches, RDS can model non-linear and non-Gaussian degradations, which are common in practical applications. This perspective provides a more comprehensive understanding of the underlying dynamics and improves the stability and accuracy of the reconstruction. We have provided an analogy in appendix B and Q4 of the general rebuttal.
>
> >Q. Outdated baselines in dehazing and deraining tasks:
>
> Our method focuses on robustness and generalization, especially in real-world applications such as MRI reconstruction, dehazing, and deraining. Achieving practical usability in these challenging tasks is a significant accomplishment. Compared to task-specific state-of-the-art methods, our approach excels with significantly reduced FLOPs and complexities (see Table 8, L. 321, P. 9). We have updated our comparisons to include very recent methods in Table 7 (L. 321, P. 9) and provided examples in the rebuttal PDF that show where these advanced methods struggle with real-world degradations.
>
> >Q. Systematic Resolution of Theoretical Insights:
>
> We appreciate the reviewer's insight that a systematic resolution of our theoretical findings requires further research, aligning with your remark that our work opens up many potential paths for future investigations. In the appendix, we have elaborated on topics such as the basin of attraction and other related issues, sharing all our findings openly. We believe our approach and findings will inspire the community to collectively explore new and exciting directions in diffusion models based on our work. We are committed to further exploring this direction, aiming to bridge generative learning and real-world problem-solving through more robust frameworks.
>
> We appreciate your constructive feedback and insightful questions. We will incorporate your suggestions to clarify the measure-preserving property and update comparisons. Your input will help us enhance the impact and clarity of our work. Thank you for your valuable input.

---

### Author Rebuttal · Authors · 2024-08-05

We thank the reviewers (APgH, J29t, z6fz) for their insightful comments and recognition of the novelty and strong theoretical foundations, comprehensive experiments and convincing results, and applicability across areas.

We greatly appreciate the recognition that D3GM opens up many potential paths for future investigations.

>Q1.Uniqueness and differences from Diffusion Bridge models.

Current diffusion theories primarily focus on image editing and translation, e.g.,  Bridging theory and diffusion theory (Table 1, L.90, P. 3), a process that does not include the perturbation of degradation factors and precise reconstruction of the underlying image. Assuming that the forward process can almost surely arrive at the target state $y$(ideal final state of the forward process and the initial state of the reverse process) is the basis for all bridge models, where in the context of our paper, $y$ is the stationary distribution $N(\mu, \tau^2 \sigma^2)$. However, the ‘almost surely’ state $y$ is never reached in finite time $T$, and the discrepancy we defined in Sec3 is the theoretical characterization of this unaddressed issue.

In this paper we provide a completely novel view on the theoretical foundation of how the *degradation* process is modeled. The result is an approach that is more in line with the original intention of the theory of diffusion with Stability and Generalizability. We chose inverse problems as a relevant application area to demonstrate our ideas but also included a variety of challenging problem settings to explore the robustness of $D^3GM$. To the best of our knowledge, no other method can handle a *diverse range* of challenging tasks like dehazing, complex-valued MRI reconstruction, super-resolution, with a *unified underlying theoretical framework*.

>Q2.Connection between the concepts.

Score-based Generative Models (SGMs)  often perform poorly in real-world scenarios. To provide a theoretical investigation of this gap, we interpret *Transitionary SGMs* as a type of process that includes random fluctuations, i.e., *Ornstein-Uhlenbeck (OU) processes*. This perspective allows us to understand the random fluctuations in image degradation as stochastic processes, providing a foundation to reinterpret SGMs as systems that evolve over time with inherent randomness with the diffusion process as a natural extension of the SDE framework involving the OU process. Such systems are known as random dynamical systems (RDS).

The *Measure-preserving property* is introduced from the perspective of RDS: The distribution can still return to the original state despite complex degradation with *stability*. Stability in RDS ensures that the system does not exhibit erratic or divergent behavior, but instead, remains predictable and consistent in the long run.

>Q3.Motivation for analysis 3 and solution 4.

We address the robustness of diffusion models for inverse problems under domain shift and concept drift (*unknown and heterogeneous degradation*).

Current SGMs approximate the degradation process as linear and monotonous, which enhances their performance in scenarios traditionally suited to linear transformation. However, this assumption leads to significant shortcomings when faced with complex, real-world data as shown in our experiments. Intuitively, this overly idealized assumption of linear restoration leads to the accumulation of discrepancies (Prop. 2, L.155) during the reverse phase of unstable diffusion sampling, resulting in deviations that prevent a stable backward process.

>Q4. Why we choose the perspective of measure-preserving dynamics of random dynamical systems.

Consider the analogy of a stretched rubber band, which naturally seeks to return to its original but does so with a lot of oscillations when released.

This elastic behavior parallels the dynamics of the OU process, where deviations from a mean state are counteracted by a restorative force, guiding the system back towards equilibrium (i.e., final state), with random perturbations.

Our process models the noise as a stochastic component that fluctuates around a stationary process and improves the OU process with RDS. Measure-preserving dynamics ensures that while the image undergoes transformations during the denoising process, the overall statistical properties remain consistent (i.e., invariant image features), which cannot be satisfied by previous approaches (Tab.~1).

>Q5. Choices of Drift and Diffusion Coefficients:

PDF file and below are the different settings and considerations, the final version will include more details and discussion.

*Cos*: The Cos schedule is preferred for its smooth transition and better handling of complex temporal dynamics, maintaining a high signal-to-noise ratio and preventing abrupt loss of original information with a uniform denoising process.

*Log*: The log schedule's initial fast noise reduction increases early epistemic uncertainty, and its need for a large $𝑘$ parameter makes it resemble a constant schedule, losing its late-stage advantages.

*Lin*: The linear schedule fails to account for varying noise complexity, causing faster degeneration of original information and potential underfitting or overfitting during sampling.

*Quad*: The quadratic schedule's aggressive early noise reduction and slow late-stage denoising result in rapid loss of details initially and insufficient denoising later, lacking general improvements.

Diffusion models require high signal-to-noise ratios to achieve fine-grained detail and spatiotemporal coherence, particularly during the reverse process. The COS schedule supports measure-preserving dynamics by providing a consistent and controlled denoising process. It reduces the risk of accumulating errors that can distort the reconstructed image.

We appreciate the reviewers' constructive feedback and are committed to addressing all the raised concerns in our camera-ready version, ensuring a rigorous and impactful camera-ready version.

---

### Author Response · Authors · 2024-08-12

Dear Reviewers,

We would like to kindly remind you that the author-reviewer discussion period will be ending soon (Aug 13 11:59pm AoE). We greatly appreciate your time and effort in reviewing our work and would be delighted to engage in further discussions to address any remaining concerns you may have. Please do not hesitate to contact us if you have any questions or require additional information.

Best regards,

Authors

---

### Decision · Program_Chairs · 2024-09-25

**Decision:**

Accept (poster)

**Comment:**

The paper studies inverse problems through the lens of diffusion models. The majority of the reviewers agrees that this is a strong submission, in particular, a novel idea, theoretically sound and well presented. Furthermore, the author's rebuttal did a good job in clarifying existing issues, leading to overall score increases. In summary, I am recommending acceptance and encourage the authors to take all comments seriously and adjust the manuscript accordingly for the camera-ready version.